# DynaMITE-RL: A Dynamic Model for Improved Temporal Meta-Reinforcement Learning

**Anthony Liang**
University of Southern California
aliang80@usc.edu

**Guy Tennenholtz**
Google Research
guytenn@google.com

**Chih-Wei Hsu**
Google Research
cwhsu@google.com

**Yinlam Chow**
Google Deepmind
yinlamchow@google.com

**Erdem Biyik**
University of Southern California
erdem.biyik@usc.edu

**Craig Boutilier**
Google Research
cboutilier@google.com

## Abstract

We introduce *DynaMITE-RL*, a meta-reinforcement learning (meta-RL) approach to approximate inference in environments where the latent state evolves at varying rates. We model episode sessions—parts of the episode where the latent state is fixed—and propose three key modifications to existing meta-RL methods: (i) consistency of latent information within sessions, (ii) session masking, and (iii) prior latent conditioning. We demonstrate the importance of these modifications in various domains, ranging from discrete Gridworld environments to continuous-control and simulated robot assistive tasks, illustrating the efficacy of DynaMITE-RL over state-of-the-art baselines in both online and offline RL settings.

## 1 Introduction

Markov decision processes (MDPs) [4] provide a general framework in reinforcement learning (RL), and can be used to model sequential decision problems in a variety of domains, e.g., recommender systems (RSs), robot and autonomous vehicle control, and healthcare [22, 21, 7, 46, 31, 5]. MDPs assume a static environment with fixed transition probabilities and rewards [3]. In many real-world systems, however, the dynamics of the environment are intrinsically tied to latent factors subject to temporal variation. While nonstationary MDPs are special instances of partially observable MDPs (POMDPs) [24], in many applications these latent variables change infrequently, i.e. the latent variable remains fixed for some duration before changing. One class of problems exhibiting this latent transition structure is recommender systems, where a user's preferences are a latent variable which gradually evolves over time [23, 26]. For instance, a user may initially have a strong affinity for a particular genre (e.g., action movies), but their viewing habits could change over time, influenced by external factors such as trending movies, mood, etc. A robust system should adapt to these evolving tastes to provide suitable recommendations. Another example is in manufacturing settings, where industrial robots may experience unobserved gradual deterioration of their mechanical components affecting the overall functionality of the system. Accurately modelling such latent transitions caused by hardware degradation can help manufacturers optimize performance, cost, and equipment lifespan.

Our goal in this work is to leverage such a temporal structure to obviate the need to solve a fully general POMDP. To this end, we propose **Dyna**mic **M**odel for **I**mproved **T**emporal **M**eta **R**einforcement

38th Conference on Neural Information Processing Systems (NeurIPS 2024).

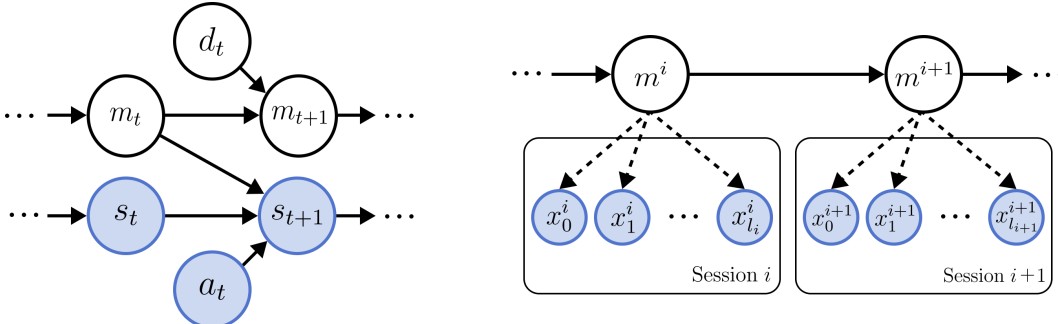

Figure 1: **(Left)** The graphical model for a DLCMDP. The transition dynamics of the environment follows $T(s_{t+1}, m_{t+1} \mid s_t, a_t, m_t)$. At every timestep $t$, an i.i.d. Bernoulli random variable, $d_t$, denotes the change in the latent context, $m_t$. Blue shaded variables are observed and white shaded variables are latent. **(Right)** A DLCMDP rollout. Each session $i$ is governed by a latent variable $m^i$ which is changing between sessions according to a fixed transition function, $T_m(m' \mid m)$. We denote $l_i$ as the length of session $i$. The state-action pair $(s_t^i, a_t^i)$ at timestep $t$ in session $i$ is summarized into a single observed variable, $x_t^i$. We emphasize that session terminations are not explicitly observed.

Learning (DynaMITE-RL), a method designed to exploit the temporal structure of sessions, i.e., sub-trajectories within the history of observations in which the latent state is fixed. We formulate our problem as a *dynamic latent contextual MDP (DLCMDP)*, and identify three crucial elements needed to enable tractable and efficient policy learning in environments with the latent dynamics captured by a DLCMDP. First, we consider consistency of latent information, by exploiting time steps for which we have high confidence that the latent variable is constant. To do so, we introduce a consistency loss to regularize the posterior update model, providing better posterior estimates of the latent variable. Second, we enforce the posterior update model to learn the dynamics of the latent variable. This allows the trained policy to better infer, and adapt to, temporal shifts in latent context in unknown environments. Finally, we show that the variational objective in meta-RL algorithms, which attempts to reconstruct the entire trajectory, can hurt performance when the latent context is nonstationary. We modify this objective to reconstruct only the transitions that share the same latent context.

Closest to our work is VariBAD [47], a meta-RL [1] approach for learning a Bayes-optimal policy, enabling an agent to quickly adapt to a new environment with unknown dynamics and reward functions. VariBAD uses variational inference to learn a posterior update model that approximates the belief over the distribution of transition and reward functions. It augments the state space with this belief to represent the agent's uncertainty during decision-making. Nevertheless, VariBAD and the Bayes-Adaptive MDP framework [35] assume the latent context is static *across an episode* and do not address settings with latent state dynamics. In this work, we focus on the dynamic latent state formulation of the meta-RL problem.

Our core contributions are as follows: (1) We introduce DynaMITE-RL, a meta-RL approach to handle environments with evolving latent context variables. (2) We introduce three key elements for learning an improved posterior update model: session consistency, modeling dynamics of latent context, and session reconstruction masking. (3) We validate our approach on a diverse set of challenging simulation environments and demonstrate significantly improved results over multiple state-of-the-art baselines in both online and offline-RL settings.

## 2 Background

We begin by reviewing relevant background including meta-RL and Bayesian RL. We also briefly summarize the VariBAD [47] algorithm for learning Bayes-Adaptive policies.

**Meta-RL.** The goal of meta-RL [1] is to quickly adapt an RL agent to an unseen test environment. Meta-RL assumes a distribution $p(\mathcal{T})$ over possible environments or *tasks*, and learns this distribution by repeatedly sampling batches of tasks during meta-training. Each task $\mathcal{T}_i \sim p(\mathcal{T})$ is described by an MDP $\mathcal{M}_i = (\mathcal{S}, \mathcal{A}, R_i, T_i, \gamma)$, where the state space $\mathcal{S}$, action space $\mathcal{A}$, and discount factor $\gamma$ are shared across tasks, while $R_i$ and $T_i$ are task-specific reward and transition functions, respectively.

The objective of meta-RL is to learn a policy that efficiently maximizes reward given a new task $\mathcal{T}_i \sim p(\mathcal{T})$ sampled from the task distribution at meta-test time. Meta-RL is a special case of a POMDP in which the unobserved variables are $R$ and $T$, which are assumed to be stationary throughout an episode.

**Bayesian Reinforcement Learning (BRL).** BRL [18] utilizes Bayesian inference to model the uncertainty of agent and environment in sequential decision making problems. In BRL, $R$ and $T$ are unknown a priori and treated as random variables with associated prior distributions. At time $t$, the *observed history* of states, actions and rewards is $\tau_{:t} = \{s_0, a_0, r_1, \ldots, r_t, s_t\}$, and the belief $b_t$ represents the posterior over task parameters $R$ and $T$ given the transition history, i.e. $b_t \triangleq p(R, T \mid \tau_{:t})$. Given the initial belief $b_0(R, T)$, the belief can be updated iteratively using Bayes' rule: $b_{t+1} = p(R, T \mid \tau_{:t+1}) \propto p(s_{t+1}, r_{t+1} \mid \tau_{:t}, R, T) \cdot b_t$. This Bayesian approach to RL can be formalized as a *Bayes-Adaptive MDP (BAMDP)* [14]. A BAMDP is an MDP over the *augmented state space* $S^+ = \mathcal{S} \times \mathcal{B}$, where $\mathcal{B}$ denotes the belief space. Given the augmented state $s_t^+ = (s_t, b_t)$, the transition function is given by $T^+(s_{t+1}^+ \mid s_t^+, a_t) = \mathbb{E}_{b_t}[T(s_{t+1} \mid s_t, a_t) \cdot \delta(b_{t+1} = p(R, T \mid \tau_{:t+1})]$, and reward function under the current belief is, $R^+(s_t^+, a_t) = \mathbb{E}_{b_t}[R(s_t, a_t)]$. The BAMDP formulation naturally resolves the exploration-exploitation tradeoff. A Bayes-optimal RL agent takes information-gathering actions to reduce its uncertainty in the MDP parameters while simultaneously maximizing the task returns. However, for most interesting problems, solving the BAMDP—and even computing posterior updates—is intractable given the continuous and typically high-dimensional nature of the task distribution.

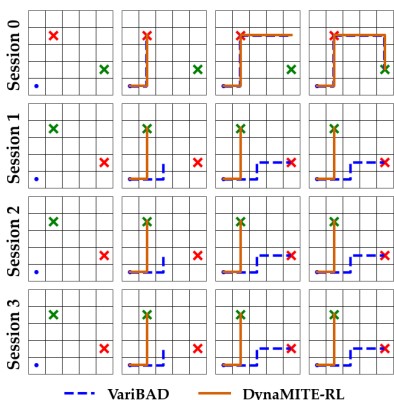

Figure 2: VariBAD does not model the latent context dynamics and fails to adapt to the changing goal location. By contrast, DynaMITE-RL correctly infers the transition and consistently reaches the rewarding cell (green cross).

**VariBAD.** Zintgraf et al. [47] approximates the Bayes-optimal solution by modeling uncertainty over the MDP parameters. These parameters are represented by a latent vector $m \in \mathbb{R}^d$, the posterior over which is $p(m \mid \tau_{:H})$, where $H$ is the BAMDP horizon. VariBAD uses a variational approximation $q_\phi(m \mid \tau_{:t})$ parameterized by $\phi$ and conditioned on the observed history up to time $t$. Zintgraf et al. [47] show that $q_\phi(m \mid \tau_{:t})$ approximates the belief $b_t$. In practice, $q_\phi(m \mid \tau_{:t})$ is represented by a Gaussian distribution $q_\phi(m \mid \tau_{:t}) = \mathcal{N}(\mu(\tau_{:t}), \Sigma(\tau_{:t}))$, where $\mu$ and $\Sigma$ are sequence models (e.g., recurrent neural networks or transformers [42]) that encode trajectories to latent statistics. The variational lower bound at time $t$ is $\mathbb{E}_{q_\phi(m|\tau_{:t})}[\log p_\theta(\tau_{:H} \mid m)] - D_{KL}(q_\phi(m \mid \tau_{:t}) \parallel p_\theta(m))$, where the first term reconstructs the trajectory likelihood $p_\theta(\tau_{:H} \mid m)$ and the second term regularizes the variational posterior to a prior distribution over the latent space, typically modeled with a standard Gaussian distribution. Importantly, the trajectory up to time $t$, i.e., $\tau_{:t}$, is used in the ELBO equation to infer the posterior belief at time $t$, which then decodes the entire trajectory $\tau_{:H}$, *including future transitions*. Given the belief state distribution $q_\phi$ of a BAMDP, the policy maps both the state and belief to actions, i.e., $\pi(a_t \mid s_t, q_\phi(m \mid \tau_{:t}))$. The BAMDP solution policy $\pi^*$ is trained, e.g., via policy gradient methods, to maximize the expected cumulative return over the task distribution: $J(\pi) = \mathbb{E}_{R,T} \left[ \mathbb{E}_\pi [\sum_{t=0}^{H-1} \gamma^t r(s_t, a_t)] \right]$.

## 3 Dynamic Latent Contextual MDPs

As a special case of a BAMDP, where the belief state is parameterized with a latent context vector (analogous to the problem formulation of VariBAD), the *dynamic latent contextual MDP (DLCMDP)* is denoted by $\langle \mathcal{S}, \mathcal{A}, \mathcal{M}, R, T, \nu_0, H \rangle$, where $\mathcal{S}$ is the state space, $\mathcal{A}$ is the action space, $\mathcal{M}$ is the *latent* context space, $R : \mathcal{S} \times \mathcal{A} \times \mathcal{M} \mapsto \Delta_{[0,1]}$ is a reward function, $T : \mathcal{S} \times \mathcal{A} \times \mathcal{M} \mapsto \Delta_{\mathcal{S} \times \mathcal{M}}$ is a transition function, $\nu_0 \in \Delta_{\mathcal{S} \times \mathcal{M}}$ is an initial state distribution, $\gamma \in (0, 1)$ is a discount factor, and $H$ is the (possibly infinite) horizon.

We assume an episodic setting in which each episode begins in a state-context pair $(s_0, m_0) \sim \nu_0$. At time $t$, the agent is at state $s_t$ and context $m_t$, and has observed history $\tau_{:t} = \{s_0, a_0, r_1, \ldots, r_t, s_t\}$.

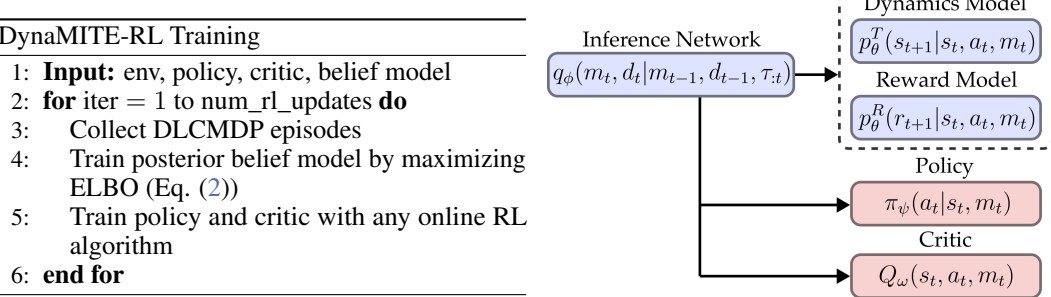

Figure 3: Pseudo-code (online RL training) and model architecture of DynaMITE-RL.

Given the history, the agent selects an action $a_t \in \mathcal{A}$, after which the state and latent context transitions according to $T(s_{t+1}, m_{t+1} \mid s_t, a_t, m_t)$, and the agent receives a reward sampled from $R(s_t, a_t, m_t)$. Throughout this process, the context $m_t$ is latent (i.e., *not observed* by the agent).

DLCMDPs embody the causal independence depicted by the graphical model in Figure 1. Particularly, DLCMDPs impose a structure on changes of the latent variable $m$, allowing the latent context $m$ to change less or more frequently. We denote by $d_t$ the random variable at which a transition occurs in $m_t$. According to Figure 1, the transition function $T$ is represented by the following factored distribution:

$$T(s_{t+1} = s', m_{t+1} = m' \mid s_t = s, a_t = a, m_t = m)$$
$$= T_s(s' \mid s, a, m)\mathbb{1}\{m' = m, d_t = 0\}T_d(d_t = 0) + \nu_0(s' \mid m')T_m(m' \mid m)\mathbb{1}\{d_t = 1\}T_d(d_t = 1),$$

where $T_m : \mathcal{M} \mapsto \mathcal{M}$ is the latent dynamics function, $T_s$ is the context-dependent state transition function, and $T_d$ is the termination probability distribution. We refer to sub-trajectories between changes in the latent context as *sessions*, which may vary in length. At the start of a new episode, a new state and a new latent context are sampled based on the distribution $\nu_0$. Each session itself is governed by an MDP parameterized with a latent context $m \in \mathcal{M}$, which changes stochastically between sessions according to the latent transition function $T_m(m' \mid m)$. For notational simplicity we use index $i$ to denote the $i^{\text{th}}$ session in a trajectory, and $m^i$ the respective latent context of that session. We emphasize that sessions switching times are latent random variables.

Notice that DLCMDPs are more general than latent MDPs [38, 29], in which the latent context is fixed throughout the entire episode; this corresponds to $d_t \equiv 0$. Moreover, DLCMDPs are closely related to POMDPs; letting $d_t \equiv 1$, a DLCMDP reduces to a general POMDP with state space $\mathcal{M}$, observation space $\mathcal{S}$, and observation function $\nu_0$. As a consequence DLCMDPs are as general as POMDPs, rendering them very expressive. Moreover, the specific temporal structure of DLCMDPs allows us to devise efficient learning algorithms that exploit the transition dynamics of the latent context, improving learning efficiency. DLCMDPs are related to DCMDPs [40], LSMDPs [8], and DP-MDP [45]. However, DCMDPs assume contexts are observed, and focus on aggregated context dynamics, LSMDPs assume that the latent contexts across sessions are i.i.d (i.e., there is no latent dynamics) and DP-MDPs assume that sessions are fixed length.

We aim to learn a policy $\pi(a_t \mid s_t, m_t)$ which maximizes the expected return $J(\pi)$ over unseen test environments. As in BAMDPs, the optimal DLCMDP Q-function satisfies the Bellman equation; $\forall s^+ \in \mathcal{S}^+, a \in \mathcal{A} : Q(s^+, a) = R^+(s^+, a) + \gamma \sum_{s^{+'} \in \mathcal{S}^+} T^+(s^{+'} \mid s^+, a) \max_{a'} Q(s^{+'}, a)$. In the following section, we present DynaMITE-RL for learning a Bayes-optimal agent in a DLCMDP.

## 4  DynaMITE-RL

We detail DynaMITE-RL, first deriving a variational lower bound for learning a DLCMDP posterior model, then outlining three principles for training DLCMDPs, and finally integrating them into our training objective.

**Variational Inference for Dynamic Latent Contexts.** Given that we do not have direct access to the transition and reward functions of the DLCMDP, following Zintgraf et al. [47], we infer the

posterior $p(m \mid \tau_{:t})$, and reason about the latent context vector $m$ instead. Since exact posterior computation over $m$ is computationally infeasible, given the need to marginalize over task space, we introduce the variational posterior $q_\phi(m \mid \tau_{:t})$, parameterized by $\phi \in \mathbb{R}^d$, to enable fast inference at every step. Our learning objective maximizes the log-likelihood $\mathbb{E}_\pi[\log p(\tau)]$ of observed trajectories. In general, the true posterior over the latent context is intractable, as is the empirical estimate of the log-likelihood. To circumvent this, we derive the *evidence lower bound (ELBO)* [27] to approximate the posterior over $m$ under the variational inference framework.

Let $\mathcal{Z} = \{m^i\}_{i=0}^{K-1}$ be the sequence of latent context vectors for $K$ sessions in an episode (note that $K$ is inherently a random variable—the exact number of sessions in an episode is not known) and $\Omega = \{d_t\}_{t=0}^{H-1}$ denote the collection of session terminations. We use a parametric generative distribution model for the state-reward trajectory, conditioned on the action sequence: $p_\theta(s_0, r_1, s_1, \ldots, r_H, s_H \mid a_0, \ldots, a_{H-1})$. In what follows, we drop the conditioning on $a_{:H-1}$ for the sake of brevity.

The variational lower bound can be expressed as:

$$\log p_\theta(\tau) \geq \underbrace{\mathbb{E}_{q_\phi(\mathcal{Z},\Omega|\tau_{:t})}\big[\log p_\theta(\tau \mid \mathcal{Z}, \Omega)\big]}_{\text{trajectory reconstruction}} - \underbrace{D_{KL}(q_\phi(\mathcal{Z}, \Omega \mid \tau_{:t}) \parallel p_\theta(\mathcal{Z}, \Omega))}_{\text{prior regularization}} = \mathcal{L}_{\text{ELBO},t}, \quad (1)$$

which can be estimated via Monte Carlo sampling over a learnable approximate posterior $q_\phi$. In optimizing the reconstruction loss of session transitions and rewards, the learned latent variables should capture the unobserved MDP parameters. The full derivation of the ELBO for a DLCMDP is provided in Appendix A.1.

Figure 2 depicts a (qualitative) didactic GridWorld example with two possible rewarding goals that alternate between sessions. The VariBAD agent does not account for latent goal dynamics and gets stuck after reaching the goal in the first session. By contrast, DynaMITE-RL employs the latent context dynamics model to capture goal changes, and adapts to the context changes across sessions.

**Consistency of Latent Information.** In the DLCMDP formulation, each session is itself an MDP with a latent context fixed across the session. This within-context stationarity means new observations can only increase the information the agent has about this context. In other words, the agent's posterior over latent contexts should gradually hone in on the true latent distribution. Although this true distribution remain unknown, this insight suggest the use of a *session-based consistency loss*, which penalizes the agent if there is no increase in information between timestep. Our consistency objective penalizes the agent when the difference between KL-divergence of the posterior to the final posterior in the session between consecutive timesteps is positive, which is the case when there is no increase in information about a session's latent context after observing a new transition. Let $d_{H-1} = 1$ and $t_i \in \{0, \ldots, H\}$ be a random variable denoting the last timestep of session $i \in \{0, \ldots, K-1\}$, i.e., $t_i = \min\{t' \in \mathbb{Z}_{\geq 0} : \sum_{t=0}^{t'} d_t = i + 1\}$. For time $t$ in session $i$, we define,

$$\delta_t = D_{KL}(q_\phi(m^i \mid \tau_{:t+1}) \parallel q_\phi(m^i \mid \tau_{:t_i})),$$

where $q_\phi(m^i \mid \tau_{:t_i})$ is the final posterior in session $i$. This measures the difference between our current belief at time $t$ to the final belief at the end of the session. Our temporal, session-based consistency objective is

$$\mathcal{L}_{\text{consistency},t} = \max\{\delta_{t+1} - \delta_t, \, 0\}.$$

Using temporal consistency to regularize inference introduces an explicit inductive bias that allows for better posterior estimation.

*Remark* 4.1. We introduce session-based consistency for DLCMDPs, though it is also relevant in single-session settings with stationary latent context. Indeed, as we discuss below, while VariBAD focuses on single sessions, it does not constrain the latent's posterior to be identical to final posterior belief. Consistency may be useful in settings where the underlying latent variable is stationary, but may hurt performance when this variable is indeed changing. Since our modeling approach allows latent context changes across sessions, incorporating consistency regularization does not generally hurt performance.

**Latent Belief Conditioning.** Unlike the usual BAMDP framework, DLCMDPs allow one to model temporal changes of latent contexts via dynamics $T_m(m' \mid m)$ across sessions. To incorporate this model into belief estimation, in addition to the history $(\tau_{:t}, d_{:t})$, we condition the posterior on the final

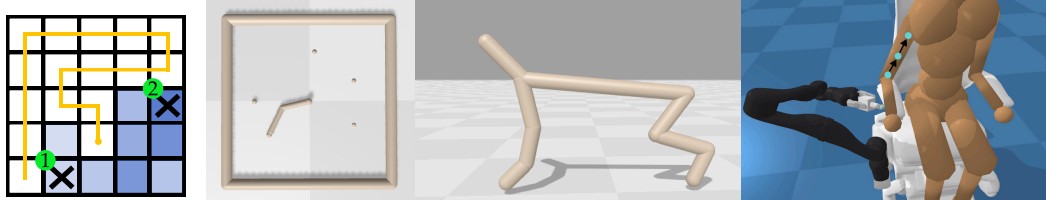

Figure 4: The environments considered in evaluating DynaMITE-RL. Each environment exhibits some change in reward and/or dynamics between sessions including changing goal locations (left and middle left), changing target velocities (middle right), and evolving user preferences of itch location (right).

latent belief $q_\phi(m', d' \mid m, d, \tau_{:t})$ from the previous session, and impose KL-divergence matching between this belief and the prior distribution $p_\theta(m' \mid m)$.

**Reconstruction Masking.** When the agent is at time $t$, Zintgraf et al. [47] encodes past interactions to obtain the current posterior $q_\phi(m \mid \tau_{:t})$ since this is all the information available for inference about the current task (see Eq. (1)). They use this posterior to decode the entire trajectory—*including future transitions*—from different sessions to optimize the lower bound during training. The insight is that decoding both the past and future allows the posterior model to perform inference about unseen states. However, we observe that when the latent context is stochastic, reconstruction over the full sequence is detrimental to training efficiency. The model is attempting to reconstruct transitions outside of the current session that may be irrelevant or biased given the latent state dynamics, rendering it a more difficult learning problem. Instead we reconstruct only the transitions within the session defined by the predicted termination indicators, i.e., at any arbitrary time $t$ within session $i$, the session-based reconstruction loss is given by

$$\mathcal{L}_{\text{session-ELBO},t} = \mathbb{E}_{q_\phi(\mathcal{Z},\Omega|\tau_{:t})}\big[\log p_\theta(\tau_{t_{i-1}+1:t_i} \mid \mathcal{Z}, \Omega)\big] - D_{KL}(q_\phi(\mathcal{Z}, \Omega \mid \tau_{:t}) \,\|\, p_\theta(\mathcal{Z}, \Omega)),$$

where $t_i$ is the last timestep of session $i$.

**DynaMITE-RL.** By incorporating the three modifications above, we obtain at the following training objective for our variational meta-RL approach:

$$\mathcal{L}_{\text{DynaMITE-RL}}(\theta, \phi) = \sum_{t=0}^{H-1} \left[ \mathcal{L}_{\text{session-ELBO},t}(\theta, \phi) + \beta \mathcal{L}_{\text{consistency},t}(\phi) \right], \tag{2}$$

where $\beta > 0$ is a hyperparameter that balances the consistency loss with the ELBO objective. We present a simplified pseudocode for online training of DynaMITE-RL in Algorithm 3a and a detailed algorithm in Appendix A.2.

**Implementation Details.** We use Proximal Policy Optimization (PPO) [37] for online RL training. We introduce a posterior inference network that outputs a Gaussian over the latent context for the $i$-th session and the session termination indicators, $q_\phi(m_{t+1}, d_{t+1} \mid \tau_{:t}, m_t, d_t)$, conditioned on the history and posterior belief from the previous session. We parameterize the inference network as a sequence model, with e.g., an RNN [9] or a Transformer [42], with different multi-layer perceptron (MLP) output heads for predicting the logits for session termination and the posterior belief. In practice, the posterior belief MLP outputs the parameters of a Gaussian distribution $q_{\phi_m}(m_{t+1} \mid \tau_{:t}, m_t) = \mathcal{N}(\mu(\tau_{:t}), \Sigma(\tau_{:t}))$ where the variance represents the agent's uncertainty about the MDP. The session termination network applies a sigmoid activation function $\sigma(x) = \frac{1}{1+e^{-x}}$ to the MLP output. Following PPO [37], the actor loss $\mathcal{J}_\pi$ and critic loss $\mathcal{J}_\omega$ are respectively given by $\mathcal{J}_\pi = \mathbb{E}_{\tau \sim \pi_\psi}[\log \pi_\psi(a \mid s, m)A(s, a, m)]$ and $\mathcal{J}_\omega = \mathbb{E}_{\tau \sim \pi_\psi}[(Q_\omega(s, a, m) - (r + V_\omega(s', m))^2]$, where $V$ is the state-value network, $Q$ is the state-action value network, and $A$ is the advantage function. We also add an entropy bonus to ensure sufficient exploration in more complex domains. A decoder network, also parameterized using MLPs, reconstructs transitions and rewards given the session's latent context $m$, current state $s_t$, and action $a_t$, i.e., $p_\theta^T(s_{t+1} \mid s_t, a_t, m_t)$ and $p_\theta^R(r_{t+1} \mid s_t, a_t, m_t)$. Figure 3b depicts the implemented model architecture. The final objective is to jointly learn the policy $\pi_\psi$, the variational posterior model $q_\phi$, and the factored likelihood model $p_\theta$ that

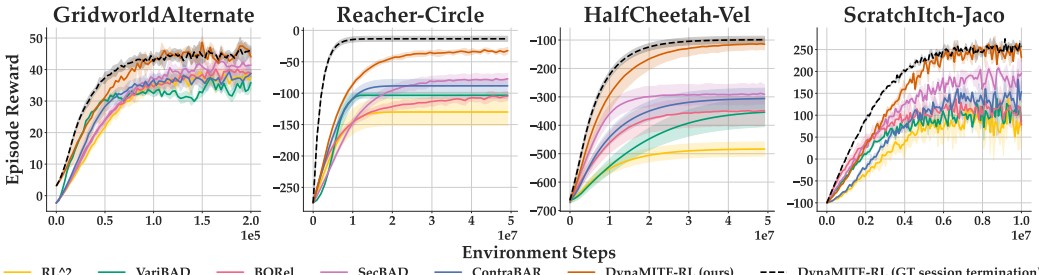

Figure 5: Learning curves for `DynaMITE-RL` and state-of-the-art baseline methods. Shaded areas represent standard deviation over 5 different random seeds for each method and 3 for ScratchItch. In each of the evaluation environments, we observe that `DynaMITE-RL` exhibits better sample efficiency and converges to a policy with better environment returns than the baseline methods.

Table 1: Average single episode returns for `DynaMITE-RL` and other state-of-the-art meta-RL algorithms across different environments. Results for all environments are averaged across 5 seeds beside ScratchItch which has 3 seeds. `DynaMITE-RL`, in bold, achieves the highest return on all of the evaluation environments and is the only method able to recover an optimal policy.

|  | Gridworld | Reacher | HC-Dir | HC-Vel | Wind+Vel | ScratchItch |
|---|---|---|---|---|---|---|
| RL$^2$ | $33.4_{\pm 1.6}$ | $-150.6_{\pm 1.2}$ | $-420.0_{\pm 8.4}$ | $-513.2_{\pm 8.7}$ | $-493.5_{\pm 1.8}$ | $50.4_{\pm 16.8}$ |
| VariBAD | $31.8_{\pm 1.9}$ | $-102.4_{\pm 4.2}$ | $-242.5_{\pm 4.8}$ | $-363.5_{\pm 3.2}$ | $-188.5_{\pm 4.4}$ | $81.8_{\pm 6.9}$ |
| BORel | $32.4_{\pm 2.4}$ | $-103.5_{\pm 4.6}$ | $-240.6_{\pm 4.3}$ | $-343.4_{\pm 3.6}$ | $-167.8_{\pm 5.4}$ | $82.5_{\pm 6.0}$ |
| SecBAD | $38.5_{\pm 3.1}$ | $-96.2_{\pm 4.8}$ | $-202.4_{\pm 10.4}$ | $-323.5_{\pm 3.4}$ | $-155.3_{\pm 5.4}$ | $101.4_{\pm 9.2}$ |
| ContraBAR | $34.5_{\pm 0.9}$ | $-101.6_{\pm 3.2}$ | $-256.5_{\pm 3.6}$ | $-312.3_{\pm 4.8}$ | $-243.4_{\pm 2.6}$ | $114.6_{\pm 24.4}$ |
| DynaMITE-RL | $\mathbf{42.9}_{\pm \mathbf{0.5}}$ | $\mathbf{-8.4}_{\pm \mathbf{5.1}}$ | $\mathbf{-68.5}_{\pm \mathbf{2.3}}$ | $\mathbf{-146.0}_{\pm \mathbf{8.1}}$ | $\mathbf{-42.8}_{\pm \mathbf{6.9}}$ | $\mathbf{231.2}_{\pm \mathbf{23.3}}$ |

minimizes the following loss:

$$\mathcal{L}(\theta, \phi, \psi) = \mathbb{E}\left[\mathcal{J}_\pi(\psi) + \lambda \cdot \mathcal{L}_{\text{DynaMITE-RL}}(\phi, \theta)\right], \tag{3}$$

where $\mathcal{J}$ is the expected return, and $\lambda > 0$ is a hyperparameter balancing the RL objective with DynaMITE-RL's variational inference objective. We also evaluate DynaMITE-RL in an offline RL setting, in which we collect an offline dataset of trajectories following an oracle goal-conditioned policy and subsequently approximate the optimal value function and RL agent using offline RL methods, e.g., IQL [28]. The value function and the policy are parameterized with the same architecture as in the online setting and will be detailed in Appendix A.5.

# 5 Experiments

We present experiments that demonstrate, while VariBAD and other meta-RL methods struggle to learn good policies given nonstationary latent contexts, DynaMITE-RL exploits the causal structure of a DLCMDP to more efficiently learn performant policies. We compare our approach to several state-of-the-art meta-RL baselines, showing significantly better evaluation returns.

**Environments.** We test DynaMITE-RL on a suite of standard meta-RL benchmark tasks including a didactic gridworld navigation, continuous control, and human-in-the-loop robot assistance as shown in Figure 4. Gridworld navigation and MuJoCo [41] locomotion tasks are considered by Zintgraf et al. [47], Dorfman et al. [12], and Choshen and Tamar [10]. We modify these environments to incorporate temporal shifts in the reward function and/or environment dynamics. To achieve good performance under these conditions, a learned policy must adapt to the latent state dynamics. More details about the environments and hyperparameters can be found in Appendix A.4 and A.5.

*Gridworld.* We modify the Gridworld environment used by Zintgraf et al. [47]. In a $5 \times 5$ gridworld, two possible goals are sampled uniformly at random in each episode. One of the two goals has a $+1$ reward while the other has $0$ reward. The rewarding goal location changes after each session

according to a predefined transition function. Goal locations are provided to the agent in the state—the only latent information is which goal has positive reward.

*Continuous Control.* We experiment with two tasks from OpenAI Gym [6]: Reacher and HalfCheetah. Reacher is a two-jointed robot arm tasked with reaching a 2D goal location that moves along a circular path according to some unknown transition function. HalfCheetah is a locomotion task which we modify to incorporate changing latent contexts w.r.t. the target direction (HalfCheetah-Dir), target velocity (HalfCheetah-Vel), and target velocity with opposing wind forces (HalfCheetah-Wind+Vel).

*Assistive Itch Scratching.* Assistive Itch Scratch is part of the Assistive-Gym benchmark [15] consisting of a human and a wheelchair-mounted 7-degree-of-freedom (DOF) Jaco robot arm. The human has limited-mobility and requires robot assistance to scratch an itch. We simulate stochastic latent context by moving the itch location—unobserved by the agent—along the human's right arm.

**Meta-RL Baselines.** We compare DynaMITE-RL to several state-of-the-art (approximately) Bayes-optimal meta-RL methods including RL$^2$ [13], VariBAD [47], BORel [12], SecBAD [8], and Contra-BAR [10]. RL$^2$ [13] is an RNN-based policy gradient method which encodes environment transitions in the hidden state and maintains them across episodes. VariBAD reduces to RL$^2$ without the decoder and the variational reconstruction objective for environment transitions. BORel primarily investigates offline meta-RL (OMRL) and proposes a few modifications such as reward relabelling to address the identifiability issue in OMRL. We evaluate the off-policy variant of BORel, trained using Soft-Actor Critic (SAC) in our DLCMDP environments. Chen et al. [8] proposes the latent situational MDP (LS-MDP), in which there is non-stationary latent contexts that are sampled i.i.d., and SecBAD, an algorithm for learning in an LS-MDP. However, they do not consider latent dynamics which a crucial aspect in many applications. ContraBAR employs a contrastive learning objective to discriminate future observations from negative samples to learn an *approximate* sufficient statistic of the history. As Zintgraf et al. [47] already demonstrate better performance by VariBAD than posterior sampling methods (e.g., PEARL [34]) we exclude such methods from our comparison.

**DynaMITE-RL outperforms prior meta-RL methods in a DLCMDP in both online and offline RL settings.** In Figure 5, we show the learning curves for DynaMITE-RL and baseline methods. We first observe that DynaMITE-RL significantly outperforms the baselines across all domains in sample efficiency and average environment returns. RL$^2$, VariBAD, BORel, SecBAD, and ContraBAR all perform poorly in the DLCMDP, converging to a suboptimal policy. VariBAD and BORel perform comparably as both share similar architecture, the only difference being the RL algorithm. By contrast, DynaMITE-RL accurately models the latent dynamics and consistently achieves high returns despite the nonstationary latent context. We also evaluate an oracle with access to ground-truth session terminations and find that DynaMITE-RL with learned session terminations effectively recovers session boundaries and matches oracle performance with sufficient training. Our empirical results validate that DynaMITE-RL learns a policy robust to changing latent contexts at inference time, while the baseline methods fail to adapt and are ultimately stuck in suboptimal behavior.

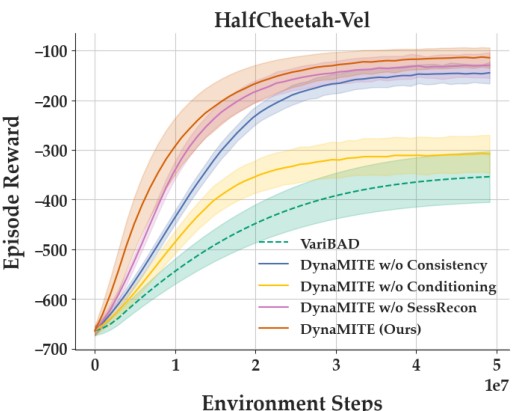

Figure 6: Ablating individual components of DynaMITE-RL. We observe that modelling latent dynamics is crucial in achieving good performance in a DLCMDP. Additionally, both consistency regularization and session reconstruction are critical for improving the sample efficiency and convergence to a better performing policy.

We further demonstrate that DynaMITE-RL outperforms BORel in an offline RL setting in Table 2 across all environments. This highlights the importance of DynaMITE-RL training objectives in learning a more accurate posterior belief model even without online environment interactions. We also experimented with a Transformer encoder to parameterize our belief model and find that a more powerful model further improves the evaluation performance.

Table 2: Average single episode returns with offline RL. Results are averaged across 5 random seeds. Algorithm with the highest average return are shown in bold. We present results for an oracle agent trained with goal information for reference.

| | Gridworld | Reacher | HC-Dir | HC-Vel | HC-Dir+Vel | ScratchItch |
|---|---|---|---|---|---|---|
| BORel | $31.4_{\pm 3.5}$ | $-102.0_{\pm 5.8}$ | $-245.0_{\pm 12.4}$ | $-354.0_{\pm 8.3}$ | $-170.0_{\pm 5.4}$ | $72.5_{\pm 4.6}$ |
| w/o Consistency | $38.2_{\pm 1.2}$ | $-33.2_{\pm 2.7}$ | $-206.0_{\pm 5.6}$ | $-212.0_{\pm 6.4}$ | $-120.0_{\pm 12.4}$ | $105.8_{\pm 8.5}$ |
| w/o Sess. Dynamics | $33.4_{\pm 1.3}$ | $-95.0_{\pm 5.2}$ | $-244.0_{\pm 6.0}$ | $-342.0_{\pm 8.6}$ | $-166.0_{\pm 9.5}$ | $74.1_{\pm 2.3}$ |
| DynaMITE-RL | $41.8_{\pm 0.6}$ | $-15.5_{\pm 3.2}$ | $-154.0_{\pm 8.6}$ | $-156.0_{\pm 4.8}$ | $-48.0_{\pm 8.6}$ | $225.5_{\pm 10.6}$ |
| w/ Transformer | $\mathbf{43.8_{\pm 0.6}}$ | $\mathbf{-8.4_{\pm 2.8}}$ | $\mathbf{-132.0_{\pm 7.4}}$ | $\mathbf{-144.0_{\pm 6.5}}$ | $\mathbf{-33.0_{\pm 5.8}}$ | $\mathbf{242.5_{\pm 7.4}}$ |
| Oracle (w/ goal) | 44.6 | $-4.8$ | $-112.0$ | $-132.2$ | $-24.4$ | 245.3 |

**Each component of DynaMITE-RL contributes to efficient learning in a DLCMDP.** We ablate the three key components of `DynaMITE-RL` to understand their impact on the resulting policy. We compare full `DynaMITE-RL` to: (i) DynaMITE-RL w/o Consistency, which does not include consistency regularization; (ii) DynaMITE-RL w/o Conditioning, which does not include latent conditioning; and (iii) DynaMITE-RL w/o SessRecon, which does not include session reconstruction. In Figure 6, we report the performance for each of these ablations and vanilla VariBAD for comparisons. First, without prior latent belief conditioning, the model converges to a suboptimal policy slightly better than `VariBAD`, confirming the importance of modeling the latent transition dynamics of a DLCMDP. Second, we find that session consistency regularization reinforces the inductive bias of changing dynamics and improves the sample efficiency of learning an accurate posterior model in DLCMDPs. Finally, session reconstruction masking also improves the sample efficiency by neglecting terms that are irrelevant and potentially biased. Similar ablation studies in the offline RL setting can be found in Table 2, reinforcing the importance of our proposed training objectives.

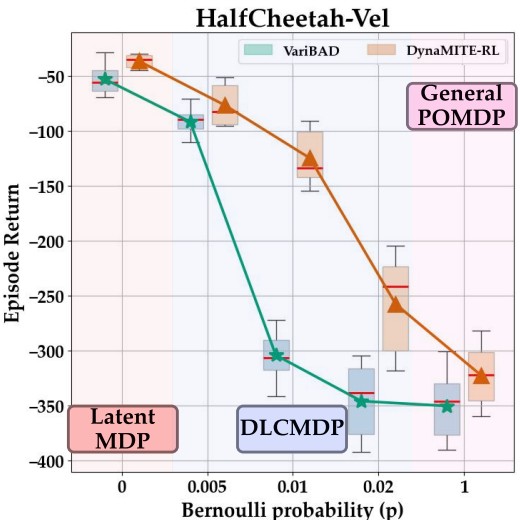

Figure 7: Ablation studies on various frequencies of latent context switches within an episode in the HalfCheetah-Vel environment. The boxplot shows the distribution over evaluation returns for 25 rollouts of trained policies with `VariBAD` and `DynaMITE-RL` . When $p = 0$, we have a latent MDP and when $p = 1$ this is equivalent to a general POMDP.

**DynaMITE-RL is robust to varying levels of latent stochasticity.** We study the effect of varying the number of latent context switches over an episode of fixed time horizon. For the HalfCheetah-Vel environment, we fix the episode horizon $H = 400$ to create multiple problems. We introduce a Bernoulli random variable, e.g $d_t \sim Bernoulli(p)$ where $p$ is a hyperparameter we set to determine the probability that the latent context changes at timestep $t$. If $p = 0$, the latent context remains unchanged throughout the entire episode, corresponding to a latent MDP. If $p = 1$, the latent context changes at every timestep, which is equivalent to a general POMDP. As shown in Figure 7, `DynaMITE-RL` performs better, on average, than `VariBAD`, with lower variance in a latent MDP. We hypothesize that, in the case of latent MDP, consistency regularization helps learn a more accurate posterior model by enforcing the inductive bias that the latent is static. Otherwise, there is no inherent advantage in modeling the latent dynamics if it is stationary.

As we gradually increase the number of context switches, the problem becomes more difficult and closer to a general POMDP. `VariBAD` performance decreases drastically because it is unable to model the changing latent dynamics while `DynaMITE-RL` is less affected, highlighting the robustness of our approach to changing latent contexts. When we set the number of contexts equal to the episode

horizon length, we recreate a fully general POMDP and again the performance between `VariBAD` and `DynaMITE-RL` converges.

# 6   Related Work

POMDPs provide a general framework modeling non-stationality and partial observability in sequential decision problems. Many model variants have been introduced, defining a rich spectrum between episodic MDPs and POMDPs. The Bayes-Adaptive MDP (BAMDP) [14] and hidden parameter MDP (HiP-MDP) [25] are both special cases of POMDPs in which environment parameters are unknown and the goal is to infer these parameters online during an episode. The BAMDP model treats unknown parameters as latent variables, which are updated based on the agent's observations, while the HiP-MDP assumes that the environment dynamics depend on hidden parameters that must be learned over time. However, neither framework addresses the dynamics of the latent parameters across sessions, but rather assumes it is constant throughout an episode.

On the other hand, models like the Latent Situational MDP (LSMDP) [8] and Dynamic Parameter MDP (DP-MDP) [44] do investigate nonstationary latent contexts. LSMDP [8] samples the latent contexts independently and identically distributed (i.i.d.) at each episode. While it introduces variability, it does not model the temporal dynamics or dependencies of these latent parameters. The DP-MDP framework addresses these dynamics by assuming that the latent parameters change at fixed intervals (fixed session lengths), making it less flexible when sessions are variable lengths. By contrast, DLCMDPs models the dynamics of the latent state and simultaneously infers *when* the transition occurs, allowing better posterior updates at inference time.

DynaMITE-RL shares conceptual similarities with other meta-RL algorithms. Firstly, optimization-based techniques [16, 11, 36] learn neural network policies that can quickly adapt to new tasks at test time using policy gradient updates. This is achieved using a two-loop optimization structure: in the inner loop, the agent performs task-specific updates where it fine-tunes the policy with a few gradient steps using the task's reward function. In the outer loop, the meta-policy parameters are updated based on the performance of these fine-tuned policies across different tasks. However, these methods do not optimize for Bayes-optimal behavior and generally exhibit suboptimal test-time adaptation. Context-based meta-RL techniques aim to learn policies that directly infer task parameters at test time, conditioning the policy on the posterior belief. Such methods include recurrent memory-based architectures [13, 43, 30, 2] and variational approaches [20, 47, 12]. VariBAD, closest to our work, uses variational inference to approximate Bayes-optimal policies. However, we have demonstrated above the limitations of VariBAD in DLCMDPs, and have developed several crucial modifications to drive effective learning a highly performant policies in our setting.

# 7   Conclusion

We developed DynaMITE-RL, a meta-RL method to approximate Bayes-optimal behavior using a latent variable model. We presented the dynamic latent contextual Markov Decision Process (DLCMDP), a model in which latent context information changes according to an unknown transition function, that captures many natural settings. We derived a graphical model for this problem setting and formalized it as an instance of a POMDP. DynaMITE-RL is designed to exploit the causal structure of this model, and in a didactic GridWorld environment and several challenging continuous control tasks, we demonstrated that it outperforms existing meta-RL methods w.r.t. both learning efficiency and test-time adaptation in both online and offline-RL settings.

There are a number of exciting directions for future research building on the DLCMDP model. While we only consider Markovian latent dynamics in this work (i.e. future latent states are independent of prior latent states given the current latent state), we plan to investigate richer non-Markovian latent dynamics. We are also interested in exploring hierarchical latent contexts in which contexts change at different timescales. Finally, we hope to extend DynaMITE-RL to other real-world applications including recommender systems (RS), autonomous driving, multi-agent coordination, etc. DLCMDPs are a good model for RS as recommender agents often interact with users over long periods of time during which the user's latent context changes irregularly, directly influencing their preferences.

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

# A    Appendix / supplemental material

## A.1    ELBO Derivation for DLCMDP

We will define a full trajectory $\tau = \{s_0, a_0, r_1, s_1, a_1, \ldots, r_{H-1}, s_H\}$ where $H$ is the horizon. $\tau_{:t}$ is the history of interactions up to a global timestep $t$, i.e. $\tau_{:t} = \{s_0, a_0, r_1, s_1, a_1, \ldots r_{t-1}, s_t\}$.

Let $\mathcal{Z} = \{m^0, \ldots, m^{K-1}\}$ be the collection of latent contexts in a trajectory where $K$ is a random variable representing the number of switches the latent variable will have until time $H$, i.e., $K = \sum_{t=0}^{H-1} d_t$. Additionally, we denote $d_t$ as the session termination prediction at timestep $t$ but $d_{H-1} \equiv 1$.

We divide a full trajectory into sessions and define a discrete random variable $t_i \in \{0, \ldots, H-1\}$ be a random variable denoting the last timestep of session $i \in \{0, \ldots, K-1\}$, i.e., $t_i = \min\{t' \in \mathbb{Z}_{\geq 0} : \sum_{t=0}^{t'} d_t = i+1\}$, with $t_{-1} \equiv -1$ . We also denote the next session index $i' = i+1$.

An arbitrary session $i'$ can then be represented as, $\{s_{t_i+1}, a_{t_i+1}, r_{t_i+1}, s_{t_i+2}, \ldots, s_{t_{i'}-1}, a_{t_{i'}-1}, r_{t_{i'}}\}$.

At any time-step $t$, we want to maximize the log-likelihood of the full dataset of trajectories, $\mathcal{D}$, collected following policy $\pi$, e.g. $\mathbb{E}_\pi[\log p_\theta(\tau)]$. However, with the presence of latent variables, whose samples cannot be observed in the training data, estimating the empirical log-likelihood is generally intractable. Instead, we optimize for the evidence lower bound (ELBO) of this function with a learned approximate posterior, $q_\phi$.

We then define the posterior inference model, $q_\phi(\mathcal{Z}, d_{:H} \mid \tau_{:t})$, which outputs the posterior distribution for the latent context and session termination predictions conditioned on the trajectory history up until timestep $t$.

Below we provide the derivation for the variational lower bound of the log-likelihood function $\log p_\theta(\tau)$ for a single trajectory:

$$
\begin{aligned}
\log p_\theta(\tau) &= \log \int_{\mathcal{Z}, \Omega} p_\theta(\tau, \mathcal{Z}, \Omega) \\
&= \log \int_{\mathcal{Z}, \Omega} p_\theta(\tau, \mathcal{Z}, \Omega) \frac{q_\phi(\mathcal{Z}, \Omega \mid \tau_{:t})}{q_\phi(\mathcal{Z}, \Omega \mid \tau_{:t})} \\
&= \log \mathbb{E}_{q_\phi(\mathcal{Z}, \Omega \mid \tau_{:t})} \left[ \frac{p_\theta(\tau, \mathcal{Z}, \Omega)}{q_\phi(\mathcal{Z}, \Omega \mid \tau_{:t})} \right] \\
&= \log \mathbb{E}_{q_\phi(\mathcal{Z}, \Omega \mid \tau_{:t})} \left[ \frac{p_\theta(\tau \mid \mathcal{Z}, \Omega)\, p_\theta(\mathcal{Z}, \Omega)}{q_\phi(\mathcal{Z}, \Omega \mid \tau_{:t})} \right] \\
&\geq \mathbb{E}_{q_\phi(\mathcal{Z}, \Omega \mid \tau_{:t})} \left[ \log p_\theta(\tau \mid \mathcal{Z}, \Omega) + \log p_\theta(\mathcal{Z}, \Omega) - \log(q_\phi(\mathcal{Z}, \Omega \mid \tau_{:t})) \right] \\
&= \underbrace{\mathbb{E}_{q_\phi(\mathcal{Z}, \Omega \mid \tau_{:t})} \left[ \log p_\theta(\tau \mid \mathcal{Z}, \Omega) \right]}_{\text{reconstruction}} - \underbrace{D_{KL}(q_\phi(\mathcal{Z}, \Omega \mid \tau_{:t})) \mid\mid p_\theta(\mathcal{Z}, \Omega))}_{\text{regularization}} \\
&= \text{ELBO}_t(\theta, \phi)
\end{aligned}
$$

We extend this to derive the lower bound for all trajectories in dataset $\mathcal{D}$.

$$
\mathbb{E}_{\tau \sim \mathcal{D}} \left[ \log p_\theta(\tau) \right] = \mathbb{E}_{\tau \sim \mathcal{D}} \left[ \mathbb{E}_{q_\phi(\mathcal{Z}, \Omega \mid \tau_{:t})} \left[ \log p_\theta(\tau \mid \mathcal{Z}, \Omega) \right] - D_{KL}(q_\phi(\mathcal{Z}, \Omega \mid \tau_{:t})) \mid\mid p_\theta(\mathcal{Z}, \Omega)) \right]
$$

**Prior**:

$$
p_\theta(\mathcal{Z}, \Omega) = p_\theta(m^0 \mid d_{:t_0}) p_\theta(d_{:t_0}) \prod_{i=0}^{K-2} p_\theta(m^{i'} \mid m^i, d_{t_i+1:t_{i'}}) p_\theta(d_{t_i+1:t_{i'}})
$$

**Variational Posterior**:

$$q_\phi(\mathcal{Z}, \Omega \mid \tau_{:t}) = q_\phi(m^0 \mid \tau_{:t_0}, d_{:t_0}) q_\phi(d_{:t_0}) \prod_{i=-1}^{K-2} q_\phi(m^{i'} \mid \tau_{t_i+1:t_{i'}}, m^i, d_{t_i+1:t_{i'}}) q_\phi(d_{t_i+1:t_{i'}})$$

**Reconstruction Term**:

$$\begin{aligned}
\log p_\theta(\tau \mid \mathcal{Z}, \Omega) &= \log p_\theta(s_0, r_1, \ldots, r_{H-1}, s_H \mid \mathcal{Z}, \Omega, a_{:H-1}) \\
&= \log \prod_{i=-1}^{K-2} p_\theta(\{(s_t, r_t)\}_{t=t_i+1}^{t_{i'}} \mid \mathcal{Z}, \Omega, a_{:H-1}) \\
&= \log \prod_{i=-1}^{K-2} \left[ p_\theta(s_{t_i+1}) \prod_{t=t_i+1}^{t_{i'}} p_\theta(s_{t+1} \mid s_t, a_t, \mathcal{Z}, d_t) \, p_\theta(r_{t+1} \mid s_t, a_t, \mathcal{Z}, d_t) \right] \\
&= \sum_{i=-1}^{K-2} \left[ \log p_\theta(s_{t_i+1}) + \sum_{t=t_i+1}^{t_{i'}} \log p_\theta(s_{t+1}, r_{t+1} \mid s_t, a_t, \mathcal{Z}, d_t) \right]
\end{aligned}$$

Putting it all together:

$$\begin{aligned}
\log p_\theta(\tau) &\geq \underbrace{\mathbb{E}_{q_\phi(\mathcal{Z},\Omega|\tau_{:t})} \left[ \log p_\theta(\tau \mid \mathcal{Z}, \Omega) \right]}_{\text{reconstruction}} - \underbrace{D_{KL}(q_\phi(\mathcal{Z}, \Omega \mid \tau_{:t}) \| p_\theta(\mathcal{Z}, \Omega))}_{\text{regularization}} \\
&= \mathbb{E}_{q_\phi(\mathcal{Z},\Omega|\tau_{:t})} \Big\{ \sum_{i=-1}^{K-2} \left[ \log p_\theta(s_{t_i+1} \mid \mathcal{Z}, d_{t_i}) + \sum_{t=t_i+1}^{t_{i'}} \log p_\theta(s_{t+1}, r_{t+1} \mid s_t, a_t, \mathcal{Z}, d_t) \right] \Big\} \\
&\quad - D_{KL}(q_\phi(m^0 \mid \tau_{:t_0}, d_{:t_0}) \| p_\theta(m^0 \mid d_{:H})) \\
&\quad - \sum_{i=0}^{K-2} D_{KL}(q_\phi(m^{i'} \mid \tau_{t_i+1:t_{i'}}, m^i, d_{t_i+1:t_{i'}}) \| p_\theta(m^{i'} \mid m^i, d_{t_i+1:t_{i'}})) \\
&\quad - \sum_{i=0}^{K-2} D_{KL}(q_\phi(d_{t_i+1:t_{i'}}) \| p_\theta(d_{t_i+1:t_{i'}}))
\end{aligned}$$

## A.2 Pseudocode for DynaMITE-RL

Here we provide the pseudocode for training DynaMITE-RL and for rolling out the policy during inference time.

---

**Algorithm 1** DynaMITE-RL

---

1: **Input:** env, $\alpha_\psi, \alpha_\omega$
2: Randomly initialize policy $\pi_\psi(a \mid s, m)$, critic $Q_\omega(s, a, m)$ decoder $p_\theta(s', r' \mid s, a, m)$, encoder $q_\phi(m' \mid \cdot)$, and replay buffer $\mathcal{D} = \emptyset$
3: **for** $i = 1$ to $N$ **do**
4:    $\mathcal{D}[i] \leftarrow$ `COLLECT_TRAJECTORY`$(\pi_\psi, q_\phi, \text{env})$
5:    ▷ Train VAE
6:    Sample batches of trajectories from $\mathcal{D}$
7:    Compute ELBO with Eq. 2 and update $\theta, \phi$
8:    ▷ Update actor and critic using PPO
9:    $\psi \leftarrow \psi - \alpha_\psi \nabla_\psi \mathcal{J}_\pi$
10:    $\omega \leftarrow \omega - \alpha_\omega \nabla_\omega \mathcal{J}_Q$
11: **end for**

---

**Algorithm 2** `COLLECT_TRAJECTORY`

---

1: **Input:** $\pi_\theta$, $q_\phi$, env
2: $(s_0, m_0) \sim \nu_0$                                         {sample initial state and belief}
3: $k = 0$                                                   {session index}
4: **for** $t = 0$ to $H - 1$ **do**
5:     $a_t \sim \pi_\psi(a_t \mid s_t, m_t)$                               {get action}
6:     $(s_{t+1}, r_{t+1}) = \text{env.step}(a_t)$                      {env step}
7:     ▷ Posterior update
8:     **if** $k == 0$ **then**
9:         $m_{t+1}, d_{t+1} \sim q_\phi(\cdot \mid \tau_{:t+1}, d_t)$
10:     **else**
11:         $m_{t+1}, d_{t+1} \sim q_\phi(\cdot \mid \tau_{:t+1}, m_{t_{k-1}}, d_t)$
12:     **end if**
13:     **if** session-terminate **then**
14:         $k \mathrel{+}= 1$                                 {increment session index}
15:         $(s_{t+1}, m_{t+1}) \sim \nu_0$                      {reset the state}
16:     **end if**
17: **end for**

---

### A.3 Additional Experimental Results

Following Zintgraf et al. [47], we measure test-time performance of meta-trained policies by evaluating per-episode return for 5 consecutive episodes, see Figure 8. DynaMITE-RL and all of the baselines are designed to maximize reward *within a single rollout* hence they generally plateau after a single episode.

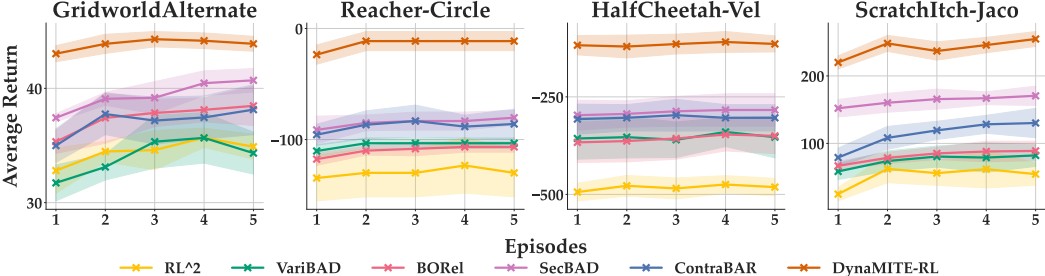

Figure 8: Average test-time performance on MuJoCo tasks and ScratchItch task, trained separately with 5 seeds for MuJoCo tasks and 3 for itching task. The meta-trained policies are rolled out for 5 episodes to show how they adapt to the task. The returns averaged across the task with 95% confidence intervals shaded. We demonstrate that in our DLCMDP setting, the baseline methods struggle to adapt to the changing dynamics of the environment while our method learns the latent transitions and achieves good performance across all domains.

The $\beta$ hyperparameter is a weight term for the consistency objective in DynaMITE-RL, which enforces an increase in information in subsequent timesteps. We run an ablation study over different values of $\beta$ for the Half-Cheetah-Vel environment in our DLCMDP setting and find that in terms of final performance, our model is robust to the different value of $\beta$.

| $\beta$ | Episode Return |
|---|---|
| 0.01 | $-69.5$ $_{\pm 2.6}$ |
| 0.1 | $-70.2$ $_{\pm 2.5}$ |
| 1 | $-68.5$ $_{\pm 2.3}$ |
| 5 | $-69.4$ $_{\pm 3.0}$ |

Table 3: Ablation study over different values of $\beta$ in the HalfCheetah-Vel environment.

## A.4 Evaluation Environment Description

In this section, we describe the details of the domains we used for our experiments. We provide visualizations of each simulation environment in Figure 4.

### A.4.1 Gridworld Navigation with Alternating Goals

Following [47], we extend the $5 \times 5$ gridworld environment as shown in Figure 2. For each episode, two goal locations are selected randomly. However, only one of the goal locations provide a positive reward when the agent arrives at the location. The rewarding goal location changes between sessions according to some transition dynamics. In our experiments, we simulate latent dynamics using a simple transition matrix: $\begin{bmatrix} 0.2 & 0.8 \\ 0.8 & 0.2 \end{bmatrix}$. Between each session, the goal location has a 20% chance of remaining the same as the previous session and 80% chance of switching to the other location. The agent receives a reward of -0.1 on non-goal cells and +1 at the goal cell, e.g.

$$r_t = \begin{cases} 1 & \text{if } s_t = g \\ -0.1 & \text{otherwise} \end{cases}$$

where $s_t$ is the current state and $g$ is the current rewarding goal cell. Similar to [47], we set the maximum episode horizon to 60 and the Bernoulli probabilty for latent context switch to 0.25 such that in expectation each episode should have 4 sessions.

### A.4.2 MuJoCo Continuous Control

For our study, we use the Brax [17] simulator, a physics engine for large scale rigid body simulation written in JAX. We use JAX [2], a machine learning framework which has just-in-time (jit) compilation that perform operations on GPU and TPU for faster training and can optimize the execution significantly. We evaluate the capacity of our method to perform continuous control tasks with high-dimensional observation spaces and action spaces.

**Reacher** is a two-joint robot arm task part of OpenAI's MuJoCo tasks [6]. The goal is to move the robot's end effector to a target 2D location. The goal locations change between each session following a circular path defined by: $[x, y] = [r cos(\alpha \cdot i), r sin(\alpha \cdot i)]$ where $i$ is the session index, $\alpha \sim \mathcal{U}(0, 2\pi)$ is the initial angle, and $r \sim \mathcal{U}(0.1, 0.2)$ is the circle's radius. The observation space is 11 dimensional consisting of information about the joint locations and angular velocity. We remove the target location from the observation space. The action space is 2 dimension representing the torques applied at the hinge joints. The reward at each timestep is based on the distance from the reacher's fingertip to the target: $r_t = -||s_f - s_g||_2 - 0.05 \cdot ||a_t||_2$ where $s_f$ is the (x, y) location of the fingertip and $s_g$ for the target location.

**Half-Cheetah** builds off of the Half-Cheetah environment from OpenAI gym [6], a MuJoCo locomotion task. In these tasks, the challenge is to move legged robots by applying torques to their joints via actuators. The state space is 17-dimensional, position and velocity of each joint. The initial state for each joint is randomized. The action space is a 6-dimensional continuous space corresponding to the torque applied to each of the six joints.

**Half-Cheetah Dir(ection):** In this environment, the agent has to run either forward or backward and this varies between session following a transition function. At the first session, the task is decided with equal probability. The reward is dependent on the goal direction:

$$r_t = \begin{cases} v_t + 0.5 \cdot ||a_t||_2 & \text{if task = forward} \\ -v_t + 0.5 \cdot ||a_t||_2 & \text{otherwise} \end{cases}$$

where $v_t$ is the current velocity of the agent.

**Half-Cheetah Vel(ocity):** In this environment, the agent has to run forward at a target velocity, which varies between sessions. The task reward is: $r_t = -||v_s - v_g||_2 - 0.05 \cdot ||a_t||_2$, where $v_s$ is the current velocity of the agent and $v_g$ is the target velocity. The second term penalizes the agent for taking large actions. The target velocity varies between session according to: $v_g = 1.5 + 1.5\sin(0.2 \cdot i)$.

**Half-Cheetah Wind + Vel:** The agent is additionally subjected to wind forces which is applied to the agent along the x-axis. Every time the agent takes a step, it drifts by the wind vector. The force is changing between sessions according to: $f_w = 10 + 10 \sin(0.3 \cdot i)$.

### A.4.3   Assistive Gym

Our assistive itch scratching task is adapted from Assistive Gym [15], similar to [39]. Assistive Gym is a simulation environment for commercially available robots to perform 6 basic activities of daily living (ADL) tasks - itch scratching, bed bathing, feeding, drinking, dressing, and arm manipulation. We extend the itch scratching task in Assistive Gym.

The itch scratching task contains a human and a wheelchair-mounted 7-DOF Jaco robot arm. The robot holds a small scratching tool which it uses to reach a randomly target scratching location along the human's right arm. The target location gradually changes along the right arm according to a predefined function, $x = 0.5 + sin(0.2 \cdot i)$ where $x$ is then projected onto a 3D point along the arm. Actions for each robot's 7-DOF arm are represented as changes in joint positions, $\mathbb{R}^7$. The observations include, the 3D position and orientation of the robot's end effector, the 7D joint positions of the robot's arm, forces applied at the robot's end effector, and 3D positions of task relevant joints along the human body. Again, the target itch location is unobserved to the agent.

The robot is rewarded for moving its end effector closer to the target and applying less than 10 N of force near the target. Assistive Gym considers a person's preferences when receiving care from a robot. For example, a person may prefer the robot to perform slow actions or apply less force on certain regions of the body. Assistive Gym computes a human preference reward, $r_H(s)$, based on how well the robot satisfies the human's preferences at state $s$. The human preference reward is combined with the robot's task success reward $r_R(s)$ to form a dense reward at each timestep, $r(s) = r_R(s) + r_H(s)$.

The full human preference reward is defined as:

$$r_H(s) = -\alpha \cdot \omega[C_v(s), C_f(s), C_{hf}(s), C_{fd}(s), C_{fdv}(s), C_d(s), C_p(s)]$$

where $\alpha$ is a vector of activations in $\{0, 1\}$ depicting which components of the preference are used and $\omega$ is a vector of weights for each preference category. $C_\bullet(s)$ is the cost for deviating from the human's preference.

$C_v(s)$ for high end effector velocities. $C_f(s)$ for applying force away from the target location. $C_{hf}(s)$ for applying high forces near the target ($> 10$ N). $C_{fd}(s)$ for spilling food or water. $C_{fdv}(s)$ for food / water entering mouth at high velocities. $C_d(s)$ for fabric garments applying force to the body. $C_p(s)$ for applying high pressure with large tools.

For our itch-scratching task, we set $\alpha = [1, 1, 1, 0, 0, 0, 0]$ and $\omega = [0.25, 0.01, 0.05, 0, 0, 0, 0]$.

## A.5   Implementation Details and Training Hyperparameters

In this section, we provide the hyperparameter values used for training each of the baselines and DynaMITE-RL. We also provide more detailed explanation of the model architecture used for each method.

### A.5.1   Online RL

We used Proximal Policy Optimization (PPO) training. The details of important hyperparameters use to produce the experimental results are presented in Table 4.

Table 4: Training hyperparameters. Dashed entries means the same value is used across all environments.

|                                          | Gridworld | Reacher | HalfCheetah | ScratchItch |
|------------------------------------------|-----------|---------|-------------|-------------|
| Max episode length                       | 60        | 400     | 400         | 200         |
| Bernoulli probability (p) for context switch | 0.07  | 0.01    | 0.01        | 0.02        |
| Number of parallel processes             | 16        | 2048    | 2048        | 32          |
| Value loss coefficient                   | 0.5       | -       | -           | -           |
| Entropy coefficient                      | 0.01      | 0.05    | 0.05        | 0.1         |
| Learning rate                            | 3e-4      | -       | -           | -           |
| Discount factor ($\gamma$)               | 0.99      | -       | -           | -           |
| GAE lambda ($\lambda_{GAE}$)             | 0.95      | -       | -           | -           |
| Max grad norm                            | 0.5       | -       | -           | -           |
| PPO clipping epsilon                     | 0.2       | -       | -           | -           |
| Latent embedding dimension               | 5         | 16      | 16          | 16          |
| Policy learning rate                     | 3e-4      | -       | -           | -           |
| VAE learning rate                        | 3e-4      | -       | -           | -           |
| State/action/reward FC embed size        | 8         | 32      | 32          | 32          |
| Consistency loss weight ($\beta$)        | 0.5       | -       | -           | -           |
| Variational loss weight ($\lambda$)      | 0.01      | -       | -           |             |

Table 5: Hyperparameters for Transformer Encoder

| Hyperparameter       | Value |
|----------------------|-------|
| Embedding dimension  | 128   |
| Num layers           | 2     |
| Num attention head   | 8     |
| Activation           | GELU  |
| Dropout              | 0.1   |

We also employ several PPO training tricks detailed in [19], specifically normalizing advantage computation, using Adam epsilon $1e - 8$, clipping the value loss, adding entropy bonus for better exploration, and using separate MLP networks for policy and value functions.

We use the same hyperparameters as above for RL$^2$ and VariBAD if applicable. For RL$^2$, the state and reward are embedded through fully connected (FC) layers, concatenated, and then passed to a GRU. The output is fed through another FC layer and then the network outputs the actions.

**ContraBAR:** Code based on the author's original implementation: https://github.com/ec2604/ContraBAR (MIT License). ContraBAR uses contrastive learning, specifically Contrastive Predictive Coding (CPC) [32], to learn an information state representation of the history. They use CPC to discriminate between positive future observations $o_{t+k}^+$ and $K$ negative observations $\{o_{t+k}^-\}_{i=1}^K$ given the latent context $c_t$. The latent context is generated by encoding a sequence of observations through an autoregressive model. They apply an InfoNCE loss to train the latent representation.

**DynaMITE-RL:** The VAE architecture consists of a recurrent encoder, which at each timestep $t$ takes as input the tuple $(a_{t-1}, r_t, s_t)$. The state, action, and reward are each passed through a different linear layers followed by ReLU activations to produce separate embedding vectors. The embedding outputs are concatenated, inputted through an MLP with 2 fully-connected layers of size 64, and then passed to a GRU to produce the hidden state. Fully-connected linear output layers generate the parameters of a Gaussian distribution: $(\mu(\tau_{:t}), \Sigma(\tau_{:t}))$ for the latent embedding $m$. Another fully-connected layer produces the logit for the session termination. The reward and state decoders are MLPs with 2 fully-connected layers of size 32 with ReLU activations. They are trained my minimizing a Mean Squared Error loss against the ground truth rewards and states. The policy and critic networks are MLPs with 2 fully-connected layers of size 128 with ReLU activations. For the

domains where the reward function is changing between sessions, we only train the reward-decoder. For HalfCheetah Wind + Vel, we also train the transition decoder.

### A.5.2 Offline RL

We use IQL [28] for offline RL training. IQL approximates the optimal value function through temporal difference learning by using expectile regression. IQL has a separate policy extraction step using advantage weighted regression (AWR) [33]. There are two main hyperparameters in IQL: $\tau \in (0, 1)$, the expectile of a random variable, and $\beta \in [0, \infty)$, an inverse temperature term for AWR. We use $\tau = 0.9$ and $\beta = 10.0$ and following [28], we use a cosine schedule for the actor learning rate. For each task, we train an oracle goal-conditioned PPO agent for data collection. The agent's initial state is randomly initialized. We collect an offline dataset of 1M environment transitions, roughly 2500 trajectories. We train IQL for 25000 offline gradient steps and report the average episode return across 5 random seeds.

### A.6 Compute Resources and Runtime

All experiments can be run on a single Nvidia RTX A6000 GPU. Implementation is written completely in JAX. The following are rough estimates of average run-time for DynaMITE-RL and each baseline method for the online RL experiments with the HalfCheetah and ScratchItch environment. These numbers vary depending on the environment; JAX-based environments (e.g. Reacher and HalfCheetah) are highly parallelized and the runtimes are orders of magnitude lower than ScratchItch. We also run multiple experiments on the same device so runtimes may be overestimated.

- $RL^2$: 4 hour, 16 hours
- VariBAD: 3 hours, 8 hours
- BORel: 3 hours, 8 hours
- SecBAD: 3 hours, 10 hours
- ContraBAR: 2.5 hours, 7 hours
- DynaMITE-RL: 3 hour, 8 hours

