# OpenReview forum: "DynaMITE-RL: A Dynamic Model for Improved Temporal Meta-Reinforcement Learning"
_NeurIPS.cc/2024/Conference — NeurIPS 2024 poster_

### Official Review · Reviewer_arsQ · 2024-07-10

**Soundness:** 2
**Presentation:** 3
**Contribution:** 2
**Rating:** 6
**Confidence:** 4

**Summary:**

The paper presents DynaMITE-RL, a meta-reinforcement learning (meta-RL) approach designed to handle environments with evolving latent states. The authors propose the Dynamic Latent Contextual Markov Decision Process (DLCMDP) model to capture the temporal structure of episodes where the latent state changes at varying rates. DynaMITE-RL incorporates three key components: session consistency, latent belief conditioning, and session reconstruction masking. The proposed method is validated through experiments in various domains, including discrete Gridworld environments, continuous-control tasks, and simulated robot assistive tasks. The results demonstrate significant improvements in policy adaptation and performance over state-of-the-art meta-RL baselines.

**Strengths:**

### S1. Novel Problem Formulation

The paper addresses a critical issue in meta-RL by introducing the DLCMDP model to capture temporal dynamics in latent states. Although this work is somewhat motivated by variBAD, this formulation is novel and provides a more realistic representation of many real-world environments where latent factors change over time.

### S2. Effective Empirical Validation

DynaMITE-RL is thoroughly validated through experiments in diverse environments. The results show significant improvements in learning efficiency and policy performance compared to existing meta-RL methods. The detailed experimental setup and comprehensive evaluation enhance the credibility of the proposed approach.

### S3. Clear Presentation

The paper is well-structured and clearly explains the proposed method and its advantages. The use of figures, such as the graphical model of DLCMDP and learning curves, effectively supports the presentation of results. The detailed pseudocode and model architecture diagrams further aid in understanding the implementation.

**Weaknesses:**

### W1. Theoretical Analysis

The paper lacks a detailed theoretical analysis of why the proposed DynaMITE-RL framework works effectively. While empirical results are strong, a deeper theoretical exploration of the underlying mechanisms and potential limitations would strengthen the paper.

### W2. Scalability Concerns

The scalability of DynaMITE-RL to more complex, large-scale environments is not thoroughly discussed. While the method shows promising results in the tested environments, a broader analysis of its scalability and practical utility in more complex scenarios is needed.

### W3. Computational Complexity

The computational complexity of the proposed method, particularly in terms of training and inference, is not explicitly addressed. Understanding the computational requirements and potential limitations in terms of resources and execution time would provide a more comprehensive evaluation of its applicability.

**Questions:**

### Q1. Scalability to Complex Environments

How does DynaMITE-RL scale to more complex, large-scale environments? Are there any specific challenges or limitations that need to be addressed for practical deployment in such scenarios?

### Q2. Computational Requirements

What are the computational requirements for training and deploying DynaMITE-RL? How does the method perform in terms of execution time and resource consumption compared to existing meta-RL methods?

**Limitations:**

The authors acknowledge the limitations related to the assumption of Markovian latent dynamics and the focus on specific benchmark environments. However, a more detailed discussion on potential negative societal impacts and strategies to mitigate them would be beneficial.

---

> ### Author Rebuttal · Authors · 2024-08-06
>
> We appreciate the reviewer’s positive feedback regarding the formulation of DLCMDP and application in many real-world scenarios compared to prior methods. We are pleased that the reviewer found the experimental setup to be “detailed” and “comprehensive” and as a result enhanced the credibility of our approach.
>
> **W1: Theoretical Analysis**
>
> We agree with the reviewer regarding the lack of more rigorous theoretical analysis. However, as the reviewer pointed out, we provide comprehensive experimental results with several state-of-the-art baselines across multiple environments highlighting the effectiveness of DynaMITE-RL in practice. More intuitively, we introduce the inductive bias of slowly evolving latent contexts and design a simple algorithm with DynaMITE-RL to exploit this making the learning problem more tractable. We are excited to incorporate more detailed theoretical analysis for our future extensions of DynaMITE-RL.
>
> **W2: Scalability to More Complex Environments**
>
> We appreciate that the reviewer is excited by future directions in extending DynaMITE-RL to complex, large-scale domains. This is certainly possible as our algorithm does not make any restrictive assumptions about the environment’s observation and action spaces. However, there are some potential challenges to deploy DynaMITE-RL in real-world environments which are interesting to consider but outside of the scope of this work.
>
> For one, in visual environments, we need to handle noisy pixel-based observations with partial information. In our simulated environments, we assume access to the full low-level environment state. One potential direction could be to use modern video architectures such as ViViT [1] and Video State Space Models [2] designed to process long image sequences to learn our belief model. Such an approach will require using larger Transformer-based architectures which have been shown to be more effective in video understanding tasks. We are eager to explore using more advanced model architectures to scale task inference to more challenging visual domains.
>
> Further, for many problems, we cannot assume access to a hand-crafted dense reward function (e.g. recommendation system). We may only have a binary signal of task success. Prior work [3] proposes an exploration bonus based on novelty to improve meta-exploration in the augmented state space which we could incorporate directly into DynaMITE-RL for sparse-reward environments. To summarize, **we believe it is possible to scale DynaMITE-RL to more complex domains and are interested in pursuing this for future work.**
>
> ***Our work is the first work that formulates this problem, proposes a simple yet effective meta-RL solution in DynaMITE-RL, and conducts proof-of-concept experiments.***
>
> **W3: Compute Resources and Runtime**
>
> We provide a description of the compute resources and runtime of DynaMITE-RL and each baseline method in Appendix A.6 which is provided in the supplementary material. We paste the text from the general response below.
>
> All experiments can be run on a single Nvidia RTX A6000 GPU. Our implementation is written completely in JAX. The following lists the average run-time for DynaMITE-RL and each baseline method for the online RL experiments with the HalfCheetah and ScratchItch environment. These numbers vary depending on the environment; JAX-based environments (e.g. Reacher and HalfCheetah) are highly parallelized and the runtimes are orders of magnitude lower than ScratchItch. We also run multiple experiments on the same device so runtimes may be overestimated.
>
> - RL2: 4 hour, 16 hours
> - VariBAD: 3 hours, 8 hours
> - BORel: 3 hours, 8 hours
> - SecBAD: 3 hours, 10 hours
> - ContraBAR: 2.5 hours, 7 hours
> - DynaMITE-RL: 3 hours, 8 hours
>
> **References:**
>
> [1] Arnab, Anurag, et al. "Vivit: A video vision transformer." Proceedings of the IEEE/CVF international conference on computer vision. 2021.
>
> [2] Li, Kunchang, et al. "Videomamba: State space model for efficient video understanding." arXiv preprint arXiv:2403.06977 (2024).
>
> [3] Zintgraf, Luisa M., et al. "Exploration in approximate hyper-state space for meta reinforcement learning." International Conference on Machine Learning. PMLR, 2021.

---

> > ### Comment · Reviewer_arsQ · 2024-08-10
> > **Response to authors rebuttal**
> >
> > I appreciate the authors for the detailed explanation about the complexity. Most of my concerns are well addressed therefore I'm raising my score.

---

> > > ### Author Response · Authors · 2024-08-13
> > >
> > > Thank you for acknowledging the content and effort of our rebuttal! We greatly appreciate you for raising the score of our paper.

---

### Official Review · Reviewer_iPsr · 2024-07-11

**Soundness:** 3
**Presentation:** 3
**Contribution:** 3
**Rating:** 7
**Confidence:** 4

**Summary:**

The authors introduce DynaMITE-RL, a meta-reinforcement learning (meta-RL) approach formulated as a dynamic latent contextual MDP (DLCMDP). This framework allows the latent context of an episode to change multiple times and at varying rates within a single episode, making it more general than both POMDPs and latent MDPs as a control framework. The authors show three criteria for an algorithm to be successful in solving a DLCMDP: consistency of latent information within sessions, session masking, and prior latent conditioning.

DynaMITE-RL addresses the limitations of existing meta-RL methods in handling dynamic latent contexts that evolve over time, which is crucial for many real-world applications. The algorithm is tested on a range of meta-RL benchmarks, including discrete Gridworld environments, continuous control tasks (Reacher, HalfCheetah), and assistive robot tasks (Assistive Itch Scratch). These environments have been altered to allow for latent context switching which can do so stochastically.

The proposed algorithm outperforms other meta-RL algorithms in the more general DLCMDP setting, in both offline and online scenarios. The results show significant improvements over state-of-the-art benchmarks in terms of training trajectories and final performance. The algorithm performs optimally when a transformer is used to encode the belief model in the offline setting.

It is shown using ablations that session consistency, latent belief conditioning, and session reconstruction masking are all important in the model's performance.

**Strengths:**

Originality

A novel meta-RL approach, DynaMITE-RL, is introduced as a DLCMDP. This is innovative as it allows the latent context of an episode to change multiple times at varying rates within a single episode, making it more general than both POMDPs and latent MDPs. This is clearly an important contribution as such context switching is evident in many real-world applications that previous methods could not handle effectively

Quality

Both the theoretical and experimental aspects of the paper bring rigour to the work and make it of high quality. The experimental setup  includes a good range of benchmarks and the ablation studies highlight the effectiveness and necessity of each of the three components of the proposed method. The results show that DynaMITE-RL outperforms state-of-the-art meta-RL algorithms

Clarity

The paper is clearly written and well-structured. The problem statement and motivation are clear, and the authors provide a detailed explanation of the DLCMDP framework and of the DynaMITE-RL algorithm. The experimental results are also clear and discussion are also clearly presented.

Significance

The DynaMITE-RL framework highlights an important set of tasks for more adaptive RL systems. This is particularly important for applications where the environment or task dynamics change over time. The ability of DynaMITE-RL to handle both online and offline settings further shows its practical relevance and applicability. The proposed approach advances the state-of-the-art in meta-RL as well as defining a novel direction for future research.

**Weaknesses:**

While hyperparameters are provided, the code itself is not. This would be relatively easy to do and would allow for very simple verification of results.

There is no discussion given to the computational complexity of the approach and how well it will scale to higher dimensional environments.

It would be useful to have an ablation study of the effects of different architectures for the belief module as well as different types of latent context dynamics.

It would be useful to have a section on real-world considerations, such as handling noisy observations or dealing with sparse rewards within this framework.

**Questions:**

Why is BORel, the meta-RL algorithm that mainly investigates offline meta-RL used in the online RL experiments?

In order to test the novel algorithm, new combinations of environments needed to be created. Why do the authors think that no such environments already exist?

**Limitations:**

Yes, in the sense that there is discussion of future work regarding non-Markovian latent dynamics and applications to real-world problems.

---

> ### Author Rebuttal · Authors · 2024-08-06
>
> We appreciate the reviewer’s positive feedback regarding the DLCMDP model and its applicability in many real-world scenarios. We are especially grateful for the reviewer’s remarks stating that our “theoretical and experimental aspects … make it [the paper] of high quality”. Moreover, the reviewer found our experiments and ablation studies to be useful in highlighting the importance of each component of DynaMITE-RL. We will provide the code for the environments and algorithms for the camera-ready version of the paper.
>
> **W1: Compute Resources and Runtime**
>
> We provide a description of the compute resources and runtime of DynaMITE-RL and each baseline method in Appendix A.6 which is included in the supplemental material.
>
> All experiments can be run on a single Nvidia RTX A6000 GPU. Our implementation is written completely in JAX. The following lists the average run-time for DynaMITE-RL and each baseline method for the online RL experiments with the HalfCheetah and ScratchItch environment. These numbers vary depending on the environment; JAX-based environments (e.g. Reacher and HalfCheetah) are highly parallelized and the runtimes are orders of magnitude lower than ScratchItch.
>
> - RL2: 4 hour, 16 hours
> - VariBAD: 3 hours, 8 hours
> - BORel: 3 hours, 8 hours
> - SecBAD: 3 hours, 10 hours
> - ContraBAR: 2.5 hours, 7 hours
> - DynaMITE-RL: 3 hours, 8 hours
>
> **W2: Ablation study of the different architectures for belief model**
>
> We agree with the reviewer that it would be interesting to have a more in-depth analysis of different architectures for the belief module. We do have some results using a Transformer-based encoder for the encoder in the belief module instead of the standard recurrent network used in prior work. We find that this does yield improvement in our offline results. However, because of computational constraints we were unable to test this in an online setting. We hypothesize that the Transformer encoder would be more beneficial in a long-horizon context in which system identification requires attending to timesteps very far in the past. We are very keen to explore the problem of belief estimation and Bayes-RL for such long-horizon applications in future work. We hope to conduct a few more experiments comparing LSTM and Transformer belief models for the camera ready paper.
>
> **W3: Real-world considerations of DynaMITE-RL**
>
> We are pleased that the reviewer is excited by future directions in extending DynaMITE-RL to complex, large-scale domains. This is certainly possible as our algorithm does not make any restricting assumptions about the environment observation and action space. However, there are some potential challenges to deploy DynaMITE-RL in real-world environments which are interesting to consider but outside of the scope of this work.
>
> For one, in visual environments, we need to handle noisy pixel-based observations with partial information. In our simulated environments, we assume access to the full low-level environment state. One potential direction could be to use modern video architectures such as ViViT [1] and Video State Space Models [2] designed to process long image sequences to learn our belief model. Such an approach will require using larger Transformer-based architectures which have been shown to be more effective in video understanding tasks. We are eager to explore using more advanced model architectures to scale task inference to more challenging visual domains.
>
> Further, for many problems, we cannot assume access to a hand-crafted dense reward function. We may only have a binary signal of task success. Prior work [3] proposes an exploration bonus based on novelty to improve meta-exploration in the augmented state space which we could incorporate directly into DynaMITE-RL for sparse-reward environments. To summarize, we believe it is possible to scale DynaMITE-RL to more complex domains and are interested in pursuing this for future work.
>
> ***Our work is the first work that formulates this problem, proposes a simple yet effective meta-RL solution in DynaMITE-RL, and conducts proof-of-concept experiments.***
>
> **Q1: Why is BORel, the meta-RL algorithm that mainly investigates offline meta-RL used in the online RL experiments?**
>
> The reviewer correctly points out that BORel [4] primarily investigates the offline approach to meta-RL. While BORel focuses on the offline-RL setting, they also propose an off-policy Soft-Actor Critic variant of their algorithm. However, we find that using an off-policy RL algorithm only provides sample-efficiency improvements over vanilla VariBAD as it allows us to reuse experience from behavior policy, but nevertheless fails to adapt to changing latent contexts in the DLCMDP environments.
>
>
> **Q2: In order to test the novel algorithm, new combinations of environments needed to be created. Why do the authors think that no such environments already exist?**
>
> To ensure fair comparison, we conduct our experiments on existing meta-RL benchmarks such as MuJoCo continuous control which have been extensively studied in prior literature. We made very minor modifications to adapt these tasks to have changing latent contexts. We further included a new itch scratching task to demonstrate a more realistic scenario where DLCMDPs may arise in the real-world. Following the reviewer's suggestion, we will look into existing open-source environments which exhibit the DLCMDP structure and include those into our experiments if possible.
>
> **References:**
>
> [1] Arnab, Anurag, et al. "Vivit: A video vision transformer." Proceedings of the IEEE/CVF international conference on computer vision. 2021.
>
> [2] Li, Kunchang, et al. "Videomamba: State space model for efficient video understanding." arXiv preprint arXiv:2403.06977 (2024).
>
> [3] Zintgraf, Luisa M., et al. "Exploration in approximate hyper-state space for meta reinforcement learning." International Conference on Machine Learning. PMLR, 2021.

---

> > ### Comment · Reviewer_iPsr · 2024-08-09
> > **Acknowledgement of receipt of response from authors**
> >
> > I thank the reviewers for these responses, and hope that with some additional comments based on this discussion it will make the paper an even stronger contribution.

---

> > > ### Author Response · Authors · 2024-08-13
> > >
> > > Thank you for acknowledging the content and effort of our rebuttal! We will incorporate the feedback from this discussion to improve the final version of our paper.

---

### Official Review · Reviewer_hpwN · 2024-07-12

**Soundness:** 2
**Presentation:** 2
**Contribution:** 2
**Rating:** 5
**Confidence:** 3

**Summary:**

The paper proposes a special variant of non-stationary MDPs, DLCMDP, where the latent context information changes according to an unknown transition function. Then the authors present DynaMITE-RL, a meta-RL approach to handle environments with evolving latent context variables. Experiments are conducted on GridWorld, two mujoco continuous control tasks, and Assistive Itch Scratch.

**Strengths:**

1. The paper targets at a more general non-stationary MDP setting than meta-RL with fixed latent context.
2. Empirical results show that the algorithm achieves better performance than other meta-RL baselines and ablation studies justify the effect of different components of the algorithm.
3. The figures and diagrams are good and makes the paper easier to understand.

**Weaknesses:**

About DLCMDPs. The definition is a little strange. I admit it's more general than HiP/latent MDP, but I would expect the dynamics to be always dependent on m - the equation after line 125 shows that when switching latent context, the state is directly sampled from a fixed initial distribution without dependency on the previous states and actions. Therefore, for now I disagree with the argument "letting dt=1, a DLCMDP reduces to a general POMDP with state space M", as POMDP's transition is always conditioned on the previous states and actions except for the initial state. Also, is there any concrete example showing that this kind of decision process exists in real world?

The algorithm section is a little vague to me. More intuitions and explanation would be helpful. Specifically, 1. Line 164, why the generative model is the probability distribution of states and rewards given the actions (is it non-stationary?), usually it's the inverse form for RL inference right? 2. How do you get $Z$ and $\Omega$ during training? If I understand correctly they are all hidden variables.

About the experiments. In table 1 and 2, although the results clearly show that the proposed methods achieve better average reward than the baselines, many of them are still negative values (e.g., HC-vel, -146.0). Is there any evidence showing that the agent indeed learn some meaningful policies for these tasks instead of some weird behaviors that cause a increase in reward? Also I would expect see results on one more domain of mujoco continuous control tasks - halfcheetah and reacher are the relatively simplest ones.

**Questions:**

See above

**Limitations:**

See above

---

> ### Author Rebuttal · Authors · 2024-08-06
>
> We appreciate the reviewer’s positive feedback and that they found the figures and diagrams helpful for understanding the paper.
>
> **W1: DLCMDP Definition**
>
> The reviewer correctly points out that according to the Equation after line 125, when the latent context changes, the next state is sampled from a fixed initial distribution which is not history dependent; in other words, there is no causal link between the final state of a session and the first state of the next session.
>
> In practice, this formulation follows prior meta-RL works where between trials, the agent is reset to the same initial state—this makes it important for the agent to maintain a belief over the latent variables across trials. However, we note **there is causal dependency between consecutive sessions through the latent context variable $m$**. We argue that our method should still work if the initial state distribution is history-dependent. The trajectory history should be fully contained in the latent $m$ because the LSTM encodes each timestep of the trajectory history to infer $m$. DLCMDPs are as general and expressive as POMDPs. POMDPs can have additional dependencies between observations, but that is not needed, and rather helps make the distinction between what is latent and a sufficient statistic, and what is observed. That said, the **specific temporal structure of DLCMDPs allows us to devise an efficient algorithm that exploits the transition dynamics of the latent context, thereby improving learning efficiency**. This, we believe, is one of the critical, and valuable, features of DLCMDPs, and an important contribution of our work.
>
> **W2: DynaMITE-RL Algorithm Clarification**
>
> In our DLCMDP setting, the latent contexts $\mathcal{Z}$ and session terminations $\Omega$ are unobserved by the agent. Following the probabilistic graphical model shown in Figure 1 of the paper, the latent variable is responsible for the generative process. Here we are not trying to learn a policy, but rather we want a model of the observable latent variables conditioned on the unobserved latent context variables. During training, the latent variable and session termination are both predicted from the hidden state of the recurrent network. At each timestep, the recurrent network encodes a tuple (state, action, reward), producing a new hidden state from which we infer the parameters of the posterior belief (Gaussian) and session termination (Bernoulli). The learning signal comes from the decoder network which predicts the ground-truth trajectory (states and rewards) conditioned on the posterior belief and termination information.
>
> During training, we first collect on-policy DLCMDP episodes. We then alternate between training the policy using any on-policy RL algorithm (e.g. PPO) and training the posterior belief model through maximize the ELBO objective derived in our paper. During inference time, we can use the trained belief model to estimate the posterior belief at each timestep given new observations and rollout the policy conditioned on the state and the current belief.
>
> **Q1: In table 1 and 2, although the results clearly show that the proposed methods achieve better average reward than the baselines, many of them are still negative values.**
>
> We emphasize that the rewards achieved by baseline in our setting are not directly comparable to those from the original paper. Under a DLCMDP, the latent dynamics causes the reward function to change between consecutive sessions. Depending on the rate of how the latent context (e.g., reward function) changes, the maximum return an agent can achieve will differ. If the latent context changes frequently, it is more difficult for the agent to infer the latent and act Bayes-optimally, consequently achieving a lower return.
>
> **Q2: Is there any evidence showing that the agent indeed learned some meaningful policies for these tasks instead of some weird behaviors that cause an increase in reward?**
>
> We have qualitative videos for each environment that highlight the differences between the resulting policies learned by DynaMITE-RL and the baseline methods. Here are links to videos of comparing agents trained with VariBAD (left) and DynaMITE-RL (right). We will include these qualitative visualizations in the supplemental material.
>
> Qualitative results: https://shorturl.at/rO917
>
> **Additional Experiment on More Complex MuJoCo Ant Walking task**
>
> Following the reviewer's suggestion, we provide additional results on the Ant task in MuJoCo. The Ant task is to navigate a four-legged ant following targets that are placed along a semi-circle path where the targets for each session change according to a predefined transition function. The action space is 8 dimensional and observation dimension is 27 dimensional representing each body part including 3D position, orientation, joint angles and joint velocities. Given the time constraints, we only able to complete experiments for VariBAD, DynaMITE-RL, and SecBAD. We provide results average over 3 seeds and 25 evaluation rollouts. We plan to provide the complete set of results for the camera-ready paper.
>
> | Method | Evaluation Return        |
> |------------|---------------|
> | VariBAD       | -80.4 $\pm$ 4.3 |
> | SecBAD        | -63.2 $\pm$ 6.2 |
> | DynaMITE-RL          | -25.6 $\pm$ 4.2 |

---

> > ### Comment · Reviewer_hpwN · 2024-08-11
> >
> > I thank the authors for the additional results and clarifications. I have increased my score to 5 correspondingly. But I'm still not fully convinced that DLCMDP is as general as POMDP.

---

> > > ### Author Response · Authors · 2024-08-13
> > >
> > > Thank you for acknowledging the content and effort of our rebuttal! We greatly appreciate you for raising the score of our paper. We will incorporate the feedback from this discussion and clarify the distinction between DLCMDPs and other variants of MDPs in the final version of the paper.

---

### Official Review · Reviewer_xD22 · 2024-07-13

**Soundness:** 3
**Presentation:** 4
**Contribution:** 3
**Rating:** 7
**Confidence:** 2

**Summary:**

This paper introduces a meta-RL method for environments with evolving latent variables. To this end, the authors introduce the notion of dynamic latent contextual MDPs, a generalization of POMDPs, which they use to model the environments. The basic idea is to have latent variables that are sampled and remain fixed throghout "sessions". The approach is evaluated on a range of tasks and is shown to outperform related state-of-the-art methods.

**Strengths:**

The paper is well written and pleasent to read.
Departures from VariBAD are well motivated and shown to be vital through the ablation study.

I like the use of color do distinguish the different methods used for comparison!

**Weaknesses:**

Somewhat incremental work.
Appendix is missing?

Minor Mistakes:
- L 146: $\max Q(s^{+'}, a')$ instead of $\max Q(s^{+'}, a)$
- L179: "remain(s)"
- L 244: There is no Figure 8, is the Appendix missing?

**Questions:**

- Figure 4: Why is the reward achieved by VariBAD on Halfcheetah-Vel worse than what is reported in the original paper?
- What do the $\Delta_{x}$ refer to on Line 113f?

**Limitations:**

Limitations are not explicitly discussed but it is not necessary here in my opinion.

---

> ### Author Rebuttal · Authors · 2024-08-06
>
> We appreciate the reviewer’s positive feedback that the paper is “well-written and pleasant to read”. We are glad that the reviewer finds the DLCMDP problem setting “well-motivated” and our extensive set of baseline comparisons and ablation studies strongly supports our technical claims. We further appreciate the reviewer for pointing out mistakes in the paper and will certainly incorporate these edits into the camera ready version.
>
> The Appendix is included in the supplemental material zip folder.
>
> **Q1: Why is the reward achieved by VariBAD on Halfcheetah-Vel worse than what is reported in the original paper?**
>
> We paste the text from the general response below. We emphasize that the rewards achieved by baseline in our setting are not directly comparable to those from the original paper. Under a DLCMDP, the latent dynamics causes the reward function to change between consecutive sessions. Depending on the rate of how the latent context (e.g., reward function) changes, the maximum return an agent can achieve will differ. If the latent context changes frequently, it is more difficult for the agent to infer the latent and act Bayes-optimally, consequently achieving a lower return.
>
> **Q2: What does the $\Delta_x$ refer to on Line 113f?**
>
> In Line 113, we intend to write, $R: \mathcal{S} \times \mathcal{A} \times \mathcal{M} \rightarrow [0, 1]$ is the reward function and $T: \mathcal{S} \times \mathcal{A} \times \mathcal{M}  \times \mathcal{S} \rightarrow [0, 1]$ is the transition kernel. We will fix this in the camera ready version.

---

> > ### Comment · Reviewer_xD22 · 2024-08-10
> >
> > I acknowledge the authors' response and feel justified to remain at my current overall rating.

---

> > > ### Author Response · Authors · 2024-08-13
> > >
> > > Thank you for acknowledging the content and effort of our rebuttal! We will incorporate the feedback from this discussion to improve the final version of our paper.

---

### Author Rebuttal · Authors · 2024-08-06

We thank all the reviewers for their constructive feedback. We appreciate positive comments regarding the novel problem formulation of slowly evolving latent context variables in DLCMDPs, clarity of writing and presentation, and strong, comprehensive empirical results of DynaMITE-RL against state-of-the-art meta-RL baselines.

The reviewers raise interesting points regarding the scalability of DynaMITE-RL to real-world problems which we are excited to explore further in future work. We will make sure to incorporate each reviewer’s feedback and clarification discussions in the camera-ready paper. Below, we reiterate a few points from the individual reviewer responses that are important to clarify.

### **On real-world scenarios that exhibit DLCMDP structure**

Many real-world applications can be modeled with a DLCMDP, from assistive robotics to recommendation systems and self-driving cars. In our paper, we extend the itch-scratching task in the Assistive Gym benchmark which demonstrates a real-world scenario in which a DLCMDP is a suitable model. Another interesting application that we are actively studying is recommendation systems. Consider a movie recommendation agent. Depending on unobserved latent factors, such as the user’s mood, their location, and environmental factors, the user might have different preferences for movie genres. For example, if the user is with their partner on a date, they might prefer romance movies while if they’re with friends they might prefer action or thriller genres. Importantly, their preferences do not change abruptly. There are multiple timesteps where the latent context remains constant (we refer to these as sessions) after which context will change gradually according to some latent dynamics.

### **Scalability of DynaMITE-RL to complex, real-world applications**

We are pleased that all reviewers are excited by future directions in extending DynaMITE-RL to complex, large-scale domains. This is certainly possible as our algorithm does not make any restrictive assumptions about the environment’s observation and action spaces. However, there are some potential challenges to deploy DynaMITE-RL in real-world environments which are interesting to consider but outside of the scope of this work.

For one, in visual environments, we need to handle noisy pixel-based observations with partial information. In our simulated environments, we assume access to the full low-level environment state. One potential direction could be to use modern video architectures such as ViViT [1] and Video State Space Models [2] designed to process long image sequences to learn our belief model. Such an approach will require using larger Transformer-based architectures which have been shown to be more effective in video understanding tasks. We are eager to explore using more advanced model architectures to scale task inference to more challenging visual domains.

Further, for many problems, we cannot assume access to a hand-crafted dense reward function. We may only have a binary signal of task success. Prior work [3] proposes an exploration bonus based on novelty to improve meta-exploration in the augmented state space which we could incorporate directly into DynaMITE-RL for sparse-reward environments. To summarize, we believe it is possible to scale DynaMITE-RL to more complex domains and are interested in pursuing this for future work.

***Our work is the first work that formulates this problem, proposes a simple yet effective meta-RL solution in DynaMITE-RL, and conducts proof-of-concept experiments.***

### **Computational Complexity and Resources**

We provide a description of the compute resources and runtime of DynaMITE-RL and each baseline method in Appendix A.6 with the supplemental material.

All experiments can be run on a single Nvidia RTX A6000 GPU. Our implementation is written completely in JAX. The following is the average run-time for DynaMITE-RL and each baseline method for the online RL experiments with the HalfCheetah and ScratchItch environments. These numbers vary depending on the environment; JAX-based environments (e.g., Reacher and HalfCheetah) are highly parallelized and the runtimes are orders of magnitude lower than ScratchItch.

- RL2: 4 hour, 16 hours
- VariBAD: 3 hours, 8 hours
- BORel: 3 hours, 8 hours
- SecBAD: 3 hours, 10 hours
- ContraBAR: 2.5 hours, 7 hours
- DynaMITE-RL: 3 hours, 8 hours

### **Negative returns in HalfCheetah experiments**

We emphasize that the rewards achieved by the baseline in our setting are not directly comparable to those from the original paper. Under a DLCMDP, the latent dynamics causes the reward function to change between consecutive sessions. Depending on the rate of how the latent context (e.g., reward function) changes, the maximum return an agent can achieve will differ. If the latent context changes frequently, it is more difficult for the agent to infer the latent and act Bayes-optimally, consequently achieving a lower return.

### **Additional Experiment on More Complex MuJoCo Ant Walking task**

Following Reviewer hpwN's suggestion, we provide additional results on the MuJoCo Ant task. The Ant task is to navigate a four-legged ant following targets that are placed along a semi-circle path. We average evaluation results over 3 random seeds and 25 rollouts.

| Method | Evaluation Return        |
|------------|---------------|
| VariBAD       | -80.4 $\pm$ 4.3 |
| SecBAD        | -63.2 $\pm$ 6.2 |
| DynaMITE-RL          | -25.6 $\pm$ 4.2 |

### **References**

[1] Arnab, Anurag, et al. "Vivit: A video vision transformer." Proceedings of the IEEE/CVF international conference on computer vision. 2021.

[2] Li, Kunchang, et al. "Videomamba: State space model for efficient video understanding." arXiv preprint arXiv:2403.06977 (2024).

[3] Zintgraf, Luisa M., et al. "Exploration in approximate hyper-state space for meta reinforcement learning." International Conference on Machine Learning. PMLR, 2021.

---

### Decision · Program_Chairs · 2024-09-25

**Decision:**

Accept (poster)

**Comment:**

This paper develops a novel meta-RL approach that handles environments with evolving latent contexts — a nice advance over existing meta-RL methods. The paper is clear, technically novel, innovative in its DLCMDP formulation, and supported by a strong empirical evaluation and ablation studies showing improvement over current meta-RL methods. There were some minor concerns with scalability/generality to more complex environments, uncertainly on the relationship between DLCMDPs and other MDP formulations such as POMDPs,  and lack of theoretical analysis, but it’s overall a solid paper with good potential impact. The authors are strongly encouraged to incorporate the reviewers’ feedback into the revision of their paper.